# CycleAug: Cycle-Consistent Visual Augmentation for Large Multimodal Models

## Abstract

Training multimodal large language models (MLLMs) requires high-quality image-question-answer (IQA) triplets, which are labour-intensive to curate and often lack diversity. We propose a novel data augmentation framework for visual instruction tuning that efficiently generates diverse synthetic images based on existing IQA anchor triplets. To ensure that the generated images align with their associated QA pairs, we propose CycleAug — *cycle-consistency visual augmentation* which involves synthesizing images from text (text → image) and then performing a verification step to confirm that the answers derived from the synthetic images match the original answers (image → text), ensuring consistency across images, questions, and answers. By combining synthetic images with high-quality real data in the training phase, we demonstrate these synthetic triplets act as an implicit regularization, which improves the robustness of MLLMs and enables analogical reasoning. Extensive experiments show that our approach improves model performance on multiple visual question-answering benchmarks without additional real-world data. This work highlights the potential of leveraging visual foundational models to enhance visual instruction tuning in MLLMs.

## 1 Introduction

Recent advancements in large language models paved the way for the success of multimodal large language models (MLLMs) (OpenAI, 2023; Liu et al., 2024b). Taking advantage of the progress achieved in large language models, connecting visual modality with LLMs is being intensively researched. Apart from the design of the model's architecture, it is universally agreed that carefully curated, high-quality image-text pairs are crucial for achieving strong performance in MLLMs (Liu et al., 2024b; Zhu et al., 2023b; Bai et al., 2024). However, the process of curating this data is labour-intensive due to the limited accessibility of image-question-answer data. To address this issue, LLaVA(Liu et al., 2024b) leverages GPT-4's language processing capabilities to generate corresponding complex visual question-answer pairs grounded on real images. Nevertheless, this approach still heavily relies on real-world images, which poses a challenge to further scaling and diversifying the training data.

The need for more diverse cross-modal data leads us to ask: *Can we efficiently generate new images based on an anchor triplet of image-question-answer?* We hypothesize that diversifying images for each QA pair could improve modality fusion, similar to analogical reasoning in cognitive science, which refers to the ability to perceive and use relational similarity between two situations or events (Gentner & Maravilla, 2017). By generating multiple diverse images asscociated with the same QA pair, the model is encouraged to focus on the visual features most relevant to the question. For instance, consider several images of an apple in different backgrounds all paired with one question: *What color is the apple in the image?*. In this scenario, the model learns to attend to the common and relevant element, the apple, ignoring irrelevant elements. This approach enhances the fine-grained correspondence between visual components and textual descriptions, thereby improving the model's ability to generalize and perform accurately on visual reasoning tasks.

Building upon these insights, we propose a framework for efficiently generating diverse multimodal datasets for visual instruction tuning of MLLMs. In this framework, a diverse set of synthetic images is generated based on multi-turn complex question-answering (QA) conversations. This approach aims to improve the model's ability to generalize and understand relational patterns in visual content.

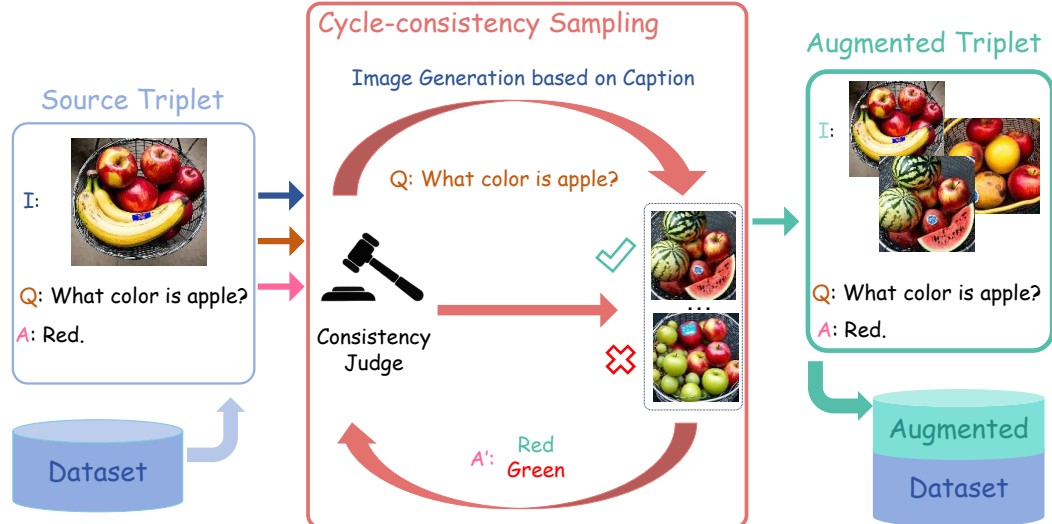

Figure 1: **Overview of our proposed method of visual augmentation for LMMs.** Figure 1 illustrates our proposed visual augmentation technique for large multimodal models (LMMs) through cycle-consistency sampling. Starting with a source triplet $(I, Q, A)$, the process generates a synthetic image(§ 3.2) and evaluates its consistency with the original response using a "Consistency Judge"(§ 3.3). If the new image and caption are deemed consistent, they are added to the augmented dataset. This approach enriches the conditional distribution of image given question-answer pairs, and we later show these samples act as an implicit regularization during instruction tuning(§ 3.4).

To ensure that the generated images accurately correspond to their associated question-answer (QA) pairs, we propose a method leveraging *cycle-consistency sampling* across images, questions, and answers. Initially, we employ a superior MLLM (LMMS Lab, 2024) to convert an anchored image—refering to the original image from our selected IQA triplet— into a detailed caption. Then, we synthesize images conditioned on this caption using a text-to-image model. To evaluate the fidelity of these images to the original QA pair, we employ MLLMs to ask the same questions about the generated images and compare the consistency of the answers with the original ones. We retain only those synthesized images that demonstrate high answer consistency, ensuring that the synthetic images are correctly associated with their QA pairs. Crucially, the success of this method depends on having a strong consistency judge to accurately assess the alignment between the generated images and their QA pairs. Without a reliable consistency check, even well-aligned images can introduce noise if subtle details, such as small objects or colors, are inaccurately represented. This judge helps bridge the gap caused by limitations in the diffusion model and caption generation, ensuring that only high-quality pairs are included in the augmented training set. Finally, we do not alter the training scheme but directly mix these synthetic pairs with the real ones, allowing the synthetic samples to serve as an implicit form of regularization during MLLM instruction tuning, which improves overall model performance.

We demonstrate the effectiveness of our CYCLEAUG through extensive experimentation. Using the same amount of real data, our approach increases the diversity of the training set while maintaining the high quality required for complex reasoning and multi-turn dialogues. Models trained with our augmented data demonstrate performance improvements on eight VQA benchmarks with different focuses, leading to a performance gain of 1.0% averaged on 13 benchmarks.

In summary, our key contributions are as follows:

- We explore the impact of synthetic images for instruction tuning of MLLMs and propose a cross-modal synthetic data generation framework using *cycle-consistency sampling*. This framework generates cross-modal training data that includes diverse synthetic images with high-quality visual question answering conversations.

- We reason that synthetic images generated using the aforementioned pipeline effectively augment the training set for instruction tuning and act as an *implicit regularization*.

- We conduct extensive experiments to validate the effectiveness of our proposed data generation method. Our approach improves model performance on multiple VQA benchmarks without the need for any additional real-world data.

## 2 RELATED WORK

**Multimodal Large Language Models.** The evolving capabilities of Large Language Models (LLMs) (Brown et al., 2020; Achiam et al., 2023; Touvron et al., 2023; Chiang et al., 2023) in managing complex tasks mark significant progress toward advanced machine intelligence. However, to achieve more generalizable systems, the integration of multimodal capabilities is essential (Baltrušaitis et al., 2018), which necessitates a thorough investigation into how LLMs can incorporate visual knowledge (Yin et al., 2023). Contrastive learning enables models like CLIP (Radford et al., 2021; Jia et al., 2021; Sun et al., 2023; Zhai et al., 2023) to align visual and textual information within a unified representation space, rendering these models as common choices for the visual encoders for Multimodal Large Language Models (MLLMs). Many approaches focus on developing various connection modules to bridge visual encoders and language models and enhance the interplay between visual and textual data. Some methods project the outputs from visual encoders into image tokens using Q-Formers (Bai et al., 2023; Li et al., 2023; Dai et al., 2024; Tong et al., 2024) and Multilayer Perceptrons (MLPs) (Mokady et al., 2021; Liu et al., 2024c;a; Zhu et al., 2024), and concatenate them with text tokens before feeding them into LLMs. Other methods fuse the visual features with text features directly with Perceiver Resamplers (Alayrac et al., 2022) and plugging in visual expert modules in Transformer layers (Wang et al., 2023b; Zhang et al., 2023a). Another line of research involves using expert models, such as image captioning models, to directly translate multimodal inputs into different languages without additional training (Guo et al., 2023; Wang et al., 2023a; Zhu et al., 2023a). In addition to architectural design, the training recipe profoundly influences the performance of MLLMs. Pretraining and instruction tuning are widely recognized as essential procedures for the models mentioned (Ouyang et al., 2022; Liu et al., 2024c;a). Recognizing the capabilities of MLLMs, we aim to enhance their performance and develop a flexible approach for generating synthetic data specifically tailored for instruction tuning, thereby supporting continuous scalability.

**Visual Question Answering (VQA).** Visual Question Answering has been a long-standing problem in the field of computer vision (Antol et al., 2015). The task is defined as answering a question in natural language based on a provided image, thus necessitating multimodality (Ren et al., 2015). Before the emergence of large-scale VLMs, methods such as bilinear models (Fukui et al., 2016; Ben-Younes et al., 2019), bottom-up attention (Jiang et al., 2020; Anderson et al., 2018), neural module networks (Hu et al., 2018; 2017; Yu et al., 2019), and transformer-based methods (Chen et al., 2020; Li et al., 2020; 2019) were designed to tackle VQA tasks. Currently, visual question answering primarily utilizes Multimodal Large Language Models (MLLMs), which offer significantly greater capabilities than the aforementioned techniques. VisualGPT (Chen et al., 2022) and Frozen (Tsimpoukelli et al., 2021) represent some of the early efforts to apply MLLMs to the challenge of visual question answering. Additionally, visual question answering is increasingly being used as a key performance indicator for MLLMs. A variety of datasets have been developed for visual question answering, such as VQA (Antol et al., 2015), which introduces open-ended questions about images, VQAv2 (Kv & Mittal, 2020), which addresses the balance and diversity of VQA, GQA (Hudson & Manning, 2019), focusing on real-world visual reasoning, Visual Genome (Krishna et al., 2017), which offers rich image annotations, TextVQA (Singh et al., 2019), targeting text-based questions about images, and Ocr-vqa (Mishra et al., 2019), which emphasizes OCR in images for question answering. Moreover, these datasets are integral in enhancing the capabilities of MLLMs, serving not only as benchmarks for instruction tuning but also as rich sources of training data, ensuring models can effectively interpret and respond to a wide array of visual and textual stimuli (Karpathy & Fei-Fei, 2015; Liu et al., 2024c;a; Su et al., 2023; Gong et al., 2023).

**Synthetic Data for MLLMs.** Multimodal Large Language Models (MLLMs) typically require extensive training data during both pretraining and finetuning phases (Radford et al., 2021; Li et al., 2021; Liu et al., 2024c). However, collecting such data can be labour-intensive and prone to bias (Paullada et al., 2021). To mitigate these challenges, numerous approaches have been developed to

generate synthetic data, which can be employed during either the pretraining or the instruction tuning stages. In the domain of Large Language Models, various instruction tuning frameworks have been developed to enhance the diversity and quality of synthetic data (Lou et al., 2023; Li et al., 2024; Zhang et al., 2023b; Yin et al., 2024; Wang et al., 2024a). In contrast to LLMs, the generation of synthetic data for MLLMs places greater emphasis on aligning images with corresponding textual descriptions. To generate well-aligned image-caption pairs for pretraining, current methods use GPT4-Vision to caption real images (Chen et al., 2023b), exploit LLaMA to generate richer text data (Ma et al., 2024), or employ diffusion models to create images from selected captions (Liu et al., 2024d). For instruction tuning, most works focus on utilizing existing high-quality datasets to construct instruction-formatted datasets (Dai et al., 2024; Chen et al., 2023a; Zhang et al., 2023a; Wang et al., 2024b; Luo et al., 2024; Xu et al., 2022). LLaVA (Liu et al., 2024c) develops LLaVA-Instruct-150k by transforming real images into textual descriptions, including captions and bounding boxes. This dataset then serves to prompt a text-only GPT-4 model, which generates new content according to specific requirements and demonstrations. However, these methods, which rely on real images, fail to address key challenges such as privacy concerns, inherent biases, and limited diversity. Conversely, SimVQA (Cascante-Bonilla et al., 2022) and SwapMix (Gupta et al., 2022) enhance VQA datasets by generating new images via feature swapping and concept perturbation, significantly boosting data diversity and complexity. Yet, these enhancements are primarily tailored for simple VQA tasks and do not sufficiently meet the needs of large-scale instruction tuning, as they fail to perform well at complex reasoning tasks and multi-turn dialogue. Our method aims to bridge existing gaps by generating diverse, high-quality images from question-answer pairs and multi-turn conversations, thereby facilitating efficient instruction tuning for more complex interactions.

## 3 METHODS

In this section, we start by reviewing the visual instruction tuning of multimodal large language models (MLLMs) in § 3.1. We present the problem formulation of data augmentation for visual instruction tuning and discuss the major challenges in § 3.2. Then in § 3.3 we propose our cycle-consistent data augmentation framework. Lastly we discuss the updated instruction tuning pipeline and the benefits of our cycle-consistent data augmentation in § 3.4.

### 3.1 PRELIMINARIES: VISUAL INSTRUCTION TUNING

Visual instruction tuning (Liu et al., 2024c;a) is an effective approach to endow MLLMs with strong reasoning and instruction-following capabilities. Given input image $\mathbf{X}_v$, we first utilize a pretrained CLIP visual encoder $g(\cdot)$ to extract visual features $\mathbf{Z}_v = g(\mathbf{X}_v)$. To bridge the gap between pre-trained visual and textual embeddings, Liu et al. (2024c) adopted a single-layer MLP parameterized by $\mathbf{W}$ and transform the visual features $\mathbf{Z}_v$ into the textual space, given by $\mathbf{H}_v = \mathbf{W}\mathbf{Z}_v$.

For each image $\mathbf{X}_v$ we generate multi-turn conversation data $(\mathbf{X}_q^1, \mathbf{X}_a^1, \ldots, \mathbf{X}_q^T, \mathbf{X}_a^T)$ where $T$ is the number of turns. The data is further organized as a sequence, by treating all answers as the assistant's response, and the instruction $\mathbf{X}_{\text{instruct}}^t$ at the $t$-th turn is

$$X_{\text{instruct}}^t = \begin{cases} \text{Randomly Choose } [\mathbf{X}_q^1, \mathbf{X}_v] \text{ or } [\mathbf{X}_v, \mathbf{X}_q^1] & t = 1 \\ \mathbf{X}_q^t & t > 1. \end{cases} \tag{1}$$

Then we perform instruction-tuning of the LLM on the prediction tokens with the auto-regressive training objective.

### 3.2 DATA AUGMENTATION FOR VISUAL INSTRUCTION TUNING

**Problem formulation.** While previous approaches (Liu et al., 2024c;a) focused on exploring GPT-assisted approaches to generate rich multi-turn conversation data, generating diverse and high-quality image data to benefit large-scale instruction tuning is largely understudied. In this work we consider a novel data augmentation approach tailored for visual instruction tuning, where we generate diverse images based on an anchor triplet of image-question-answer. Besides improving the richness of the set of image data for training, the generated data will also enable analogical reasoning (Gentner & Maravilla, 2017), with the availability of multiple images for each question-answer pair.

Q: Are there doors?

A: No.

≫Caption for direct sampling:
There are no doors in this image.

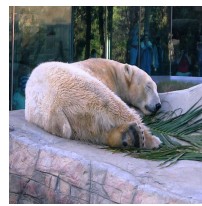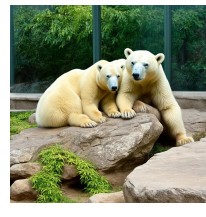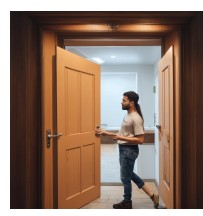

Q: What are the weather conditions today?

A: Clear.

Q: Is the sky blue and clear?

A: Yes.

≫Caption for direct sampling:
A clear blue sky on a sunny day.

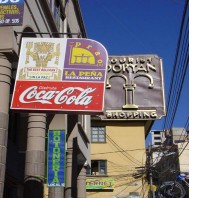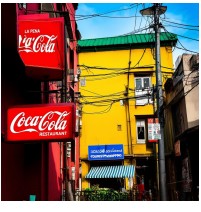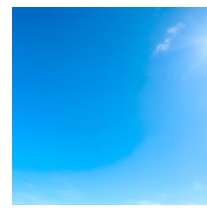

Anchor Image     Cycle-Consistency Sampling     Direct Sampling

Figure 2: **The limitations of direct sampling** from $p(I|q,a)$ stem from three reasons: (1) Empirically we found SoTA text-to-image generation models cannot handle negation words well, like "no" in the above example. (2) In negative cases, the reconstructed image should include real objects—such as the glass window in the top anchor image—rather than any hallucinated object, such as a door, which was mentioned in the question but correctly excluded in the response. (3) While CYCLEAUG maintains more details and diversifies anchored images, direct sampling is based on objects mentioned in QA, which leads to an object-oriented and homogeneous generation. The caption for CYCLEAUG could be found in Appendix§ D

**Conditional image generation.** Given the problem formulation above, a straightforward approach is to explore pretrained conditional image generation. With an image-question-answer anchor triplet $(i, q, a)$, we generate new images $\{i_k\}_{k=1}^{N}$ by sampling $i_k$ from the conditional probability distribution given by

$$p(I \mid Q = q, A = a). \tag{2}$$

At first, we ask a large language model to transform question-answer pairs into descriptive statements and subsequently use them as prompts to powerful text-conditioned image generation models (stabilityai, 2024). However, preliminary results show that synthetic images generated by this method contain undesirable contents and directly sampling from $p(i \mid q, a)$ leads to degraded image diversity, as demonstrated in Figure 2. This is mainly due to two key limitations of state-of-the-art (SoTA) diffusion models: (1) Even SoTA models cannot handle negation words well, such as "no" and "not". As a result, an image containing a door may be generated, even when the caption explicitly includes a negation (see top example in Figure 2). (2) Directly sampling from a caption transformed from the question-answer pair $(q, a)$ leads to homogeneous contents, lacking the variety in real-world images (see bottom example in Figure 2). In contrast, we show that with CYCLEAUG, we can generate diverse images consistent with the anchor triplet, which forms new and valuable triplets that enable effective data augmentation for visual instruction tuning. We also refer the readers to Section 4.3 where we run ablation study experiments to underscore the limitations of direct conditional generation. We also provides more qualitative comparison results in Appendix§ D.

### 3.3 IMAGE SYNTHESIS BY SAMPLING WITH CYCLE-CONSISTENCY

To enable effective data augmentation for visual instruction tuning, we propose CYCLEAUG, an effective data augmentation framework that samples diverse images given an anchor image-question-answer triplet, *i.e.*, from $p(I \mid Q = q, A = a)$. Our CYCLEAUG consists of two modules, a diverse image synthesis module (see Figure 3) and a cycle-consistent filtering module to ensure the generated images are compatible with the given question-answer pair (see Figure 4).

**Image synthesis module.** The diverse image synthesis module builds on an encoder-decoder architecture and generates content that is diverse yet relevant to the original image by leveraging text as

Original Images                                                    Synthetic Images

MLLMs for Image Captioning → Captions → Diffusion Models

"Describe the image in detail for reconstruction."

Figure 3: **Synthetic Image Generation.** The proposed approach utilizes an encoder-decoder structure to generate diverse images from conditional distributions, leveraging text as an information bottleneck. This contrasts with traditional methods that sample directly from the real image distribution, highlighting the improved sampling efficiency and the intentional loss of certain details such as objects' shape and location in the images to enhance diversity in the generated samples.

the interface. Specifically, we adopt a pretrained MLLM (LMMS Lab, 2024) as the encoder and obtain a detailed text description of the original image. Then we use Stable-Diffusion-3-Medium (stabilityai, 2024) as the decoder and sample images conditioned on this text description. Note that our diverse image synthesis module operates in a zero-shot manner, as opposed to previous works which finetuned the encoder-decoder model with reconstruction loss (Wu et al., 2017; Wei et al., 2024). This design enables our synthesis module to generate diverse contents, appearances, and layouts that are relevant to the original image, exploring a much broader distribution of image data. To further introduce randomness, we created a set of prompts for captioning and randomly selected a prompt from this set for each caption generation.(Appendix§ D)

**Cycle-consistency sampling.** Although by design, the image synthesis module would produce diverse images with relevant contents using texts as interface, in some cases the synthetic images may violate the question-answer pair from the anchor triplet, *e.g.* when the caption doesn't specify the attribute mentioned in the question-answer pair. To ensure consistency between the diverse synthetic images and the anchor triplet, we propose a cycle-consistency sampling module. Given anchor triplet $(i, q, a)$, the generated image $i'$ is a good synthetic sample if question $q$ yields the same answer $a$ from the anchor triplet given image $i'$, as it does from the original image $i$. In practice, we prompt a pretrained MLLM (Liu et al., 2024b) with image $i'$ and question $q$ and check if the predicted answer $a'$ matches the original answer $a$, using character-wise matching and cosine similarity in text embedding space for single word and longer answers. Crucially, we reject $i'$ if the predicted answer $a'$ does not match the true answer $a$.

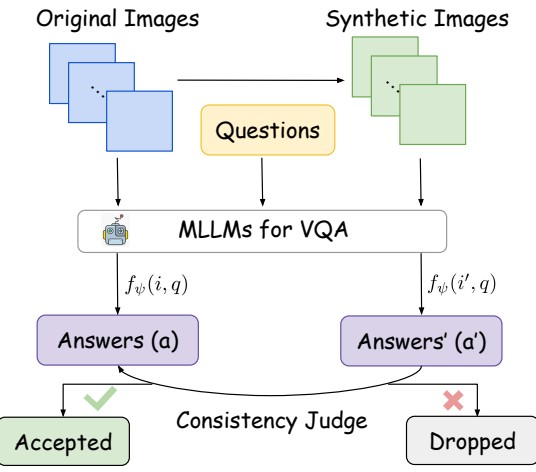

Figure 4: **Cycle-consistency Sampling.** For synthetic images generated from the same question-answer pairs, we pose identical questions to both real and synthetic images and evaluate semantic distance between the corresponding answers and reject samples that exhibit significant discrepancies. In this way, well-aligned synthetic image and question-answer triplets are selected.

**Interpreting CYCLEAUG as an implicit Bayesian sampling.** Our cycle-consistent synthetic data generation framework exploits pretrained MLLMs and diffusion models and presents an efficient approach to generate diverse and high-quality images given an anchor triplet of image-question-answer. We show the validity of our approach by showing that synthetic data generated by our framework approximates the conditional image generation method in § 3.2 with sufficient sampling. Following the results in Lemma 1, our cycle-consistency sampling filters images that violate $f_\psi(i', q) \approx a$, and by repeatedly drawing samples $i'$ from the generative distribution $p(I \mid Q = q)$, the retained set of images will accurately reflect the posterior distribution $p(I \mid Q = q, A = a)$. In our implementation, we decide to accept or reject by thresholding the cosine similarity of the

SBERT embedding of $a' := f_\psi(i', q)$ and $a$. If $S_C(a', a) \geq T$, where $S_C(\mathbf{x}, \mathbf{y}) = \frac{\langle \mathbf{x}, \mathbf{y} \rangle}{\|x\| \|y\|}$ is the cosine similarity of $\mathbf{x}, \mathbf{y}$, we accept $i'$ and otherwise reject it.

**Lemma 1** *The posterior distribution of conditional image generation can be rewritten as*

$$p(I = i \mid Q = q, A = a) = \frac{p(a \mid i, q)p(i|q)}{p(a \mid q)} = \frac{\delta(a - f_\psi(i, q))p(i|q)}{\int_{\{i|f_\theta(i,q) \approx a\}} p(i|q) \, di} \tag{3}$$

*where $f_\psi$ is a pretrained MLLM parameterized by $\psi$ and $\delta(\cdot)$ is the Dirac Delta function that approximates $p(a \mid I = i, Q = q) \approx \delta(a - f_\psi(i, q))$.*

To sample images from the posterior distribution $p(I \mid q, a)$, we start by drawing samples $i'$ directly from the proposal distribution $p(I|q) \approx p(I'|q)$ (See details in Appendix § C). For each sample $i'$, we then compute $a' = f_\psi(i', q)$. If $a'$ does not match the observed value $a$, we reject the sample $i'$. Indicated by Lemma 1, by repeating this process with a large number of samples, the retained set of images will eventually reflect the posterior distribution $p(i \mid q, a)$.

$$a' = f_\psi(i', q) = f_\psi(i, q) = a, \forall i' \sim p(I|q, a) \tag{4}$$

## 3.4 Visual Instruction Tuning with Cycle-Consistent Samples

We follow the visual instruction tuning framework in Liu et al. (2024c) and augment the instruction tuning dataset $(X_v, X_q, X_a) \in \mathcal{D}$ with cycle-consistent data augmentation samples $\{(\hat{X}_{v,k}, X_q, X_a)\}_{k=1}^N$. The new instruction following sequence data is then structured as follows:

$$X_{\text{instruct}}^t = \begin{cases} \text{Randomly Choose } [X_q^1, X_v] \text{ or } [X_v, X_q^1] \text{ or } [X_q^1, \hat{X}_{v,k}] \text{ or } [\hat{X}_{v,k}, X_q^1] & t = 1 \\ X_q^t & t > 1 \end{cases} \tag{5}$$

**Benefits of Cycle-Consistent Samples.** Visual instruction tuning with cycle-consistent data augmentation offers two key benefits: (i) Diverse, high-quality synthetic images enrich the dataset, boosting MLLM robustness. (ii) Multiple images per question-answer pair enable analogical reasoning, enhancing correspondence between visual and textual elements. As shown in Lemma 2, the training objective includes a regularization term $R$, encouraging consistent answers for original and cycle-consistent image-question pairs. This promotes robustness via adversarial-style training Goodfellow et al. (2014) and strengthens fine-grained alignment by focusing on shared concepts.

**Lemma 2** *Given image-question-answer triplets $(i, q, a)$ sampled from training data $\mathcal{D}$, the training objective of standard visual instruction tuning (Liu et al., 2024c) is given by*

$$\mathcal{L} = \mathbb{E}_{(i,q,a) \sim \mathcal{D}} \left[ \ell(f_\theta(i, q), a) \right] \tag{6}$$

*in which $\ell(\cdot)$ is a general form of loss objectives used to reduce the distance between $f_\theta(i, q)$ and $a$ during optimization.*

*Then we can rewrite the training objective of visual instruction tuning with cycle-consistent samples as the sum of the standard training objective $\mathcal{L}$ and a regularization term $R$:*

$$\mathcal{L}_{cycle\text{-}consistent} = \mathbb{E}_{(q,a) \sim p(Q,A)} \left[ \mathbb{E}_{i' \sim p(I|q,a)} \left[ \ell(f_\theta(i', q), a) \right] \right]$$
$$= \underbrace{\mathbb{E}_{(i,q,a) \sim \mathcal{D}} \left[ \ell(f_\theta(i, q), a) \right]}_{\mathcal{L}} + \underbrace{\mathbb{E}_{(i,q,a) \sim \mathcal{D}} \left[ \mathbb{E}_{i,i' \sim p(I|Q=q, A=a)} \left[ d(i, i'|q, a) \right] \right]}_{R} \tag{7}$$

*where $d(i, i' \mid q, a) = |\ell(f_\theta(i', q), a) - \ell(f_\theta(i, q), a)|$ defines a semantic distance between $f_\theta(i', q)$ and $f_\theta(i, q)$ with $a$ as an anchor in between.*

## 4 Experiments

In this section, we present the results of our method, demonstrating that it effectively improves the performance of MLLMs on general VQA tasks without utilizing additional real data. We begin by detailing our experimental setups and then showcase the results of LLaVA-1.5 fine-tuned with synthetic images sampled from the proposed conditional distribution $p(I|Q = q, A = a)$.

## 4.1 EXPERIMENTAL SETUP

We utilize the *LLama3-LLaVA-NeXT-8B* (LMMS Lab, 2024) as image captioning model, incorporating the LLaVA-NeXT (Liu et al., 2024b) framework with the LLama3-8B (Dubey et al., 2024). For image synthesis, we employ the Stable-Diffusion-3-Medium (stabilityai, 2024), which integrates the T5 text encoder (Raffel et al., 2020) capable of handling long prompts (up to 512 tokens) to ensure comprehensive image descriptions.

For finetuning multimodal large language models (MLLMs), we follow the LLaVA-1.5 (Liu et al., 2024a) data preparation and training schedules for pretraining and instruction tuning. We adopt the Vicuna-v1.5-7B (Chiang et al., 2023) language model, leveraging the LLaMA2 codebase (Touvron et al., 2023). The pre-trained CLIP ViT-L/14 (Radford et al., 2021; Dosovitskiy et al., 2021) with a $336 \times 336$ input resolution, generating $576$ visual tokens, is used as the vision encoder. Using the LLaVA framework (Liu et al., 2024a), we connect the frozen CLIP vision encoder and Vicuna LLM, training the entire LLM with the projector instead of employing parameter-efficient fine-tuning. All experiments are conducted on a machine with $8\times$ Nvidia RTX 6000 Ada GPUs. Due to invalid image links in the instruction tuning dataset, all LLaVA-1.5 scores in our analysis are reproduced to ensure fair comparisons under consistent experimental settings.

To comprehensively evaluate our method, we use 13 benchmarks for MLLM evaluation: GQA (Hudson & Manning, 2019) and VQA-v2 (Goyal et al., 2017) test visual perception with open-ended answers; MME (Fu et al., 2023) evaluates yes/no visual questions; ScienceQA (Lu et al., 2022) tests zero-shot scientific question answering; TextVQA (Singh et al., 2019) focuses on text-rich images; MMBench (Liu et al., 2023) and MMBench-CN (Liu et al., 2023) assess robustness under answer shuffling; MM-Vet (Yu et al., 2023) evaluates visual conversation skills. Metrics are computed using official implementations for consistency§ 3.3. Latency measures time until the first answer token is generated. MME scores are normalized by dividing by 2000.

## 4.2 MAIN RESULTS

In this section, we present the results of training a multimodal language model using synthetic images generated through cycle-consistency (§ 3). Due to limitations of the Stable Diffusion model in rendering text, we focused on augmenting GQA and COCO datasets instead of text-heavy datasets like OCRVQA or TextCaps. From these datasets, we generated 273,144 triplets and selected 178,304 samples using a consistency judge.

We evaluated multimodal capabilities across 13 benchmarks. As shown in Tab. 1, CYCLEAUG outperforms two baselines, achieving the highest average score of 62.32 and excelling in 8 out of 13 benchmarks. Notably, CYCLEAUG surpasses the 2-epoch baseline in accuracy (+0.6%) while requiring 73% fewer additional iterations, reducing training time from 30.2 hours to 20.1 hours. These results highlight both the effectiveness and efficiency of CYCLEAUG in leveraging synthetic data, aligning with discussions on the importance of data quality (Gadre et al., 2024).

| Method | #Iteration | $VQA^T$ | MMMU | GQA | MMVet | SQA | MME | POPE | MMB | $MMB^{CN}$ | VQAv2 | $LLaVA^w$ | VizWiz | $SEED^I$ | Avg. |
|---|---|---|---|---|---|---|---|---|---|---|---|---|---|---|---|
| baseline | 5196 | **58.2**$_{.20}$ | 35.3$_{.90}$ | 62.0$_{.50}$ | 31.1$_{1.0}$ | 66.8$_{.60}$ | **1511**$_{13}$ | 85.9$_{.20}$ | 64.3$_{.90}$ | 58.3$_{.90}$ | 78.5$_{.40}$ | 65.4$_{1.3}$ | 50.0$_{.60}$ | 66.4$_{.30}$ | 61.37$_{.27}$ |
| baseline w/ 2epoch | 10396 | 57.0$_{.46}$ | 35.1$_{.11}$ | 63.3$_{.23}$ | **32.4**$_{1.5}$ | 70.8$_{.34}$ | 1489$_{12}$ | **87.4**$_{.33}$ | 63.0$_{.23}$ | 57.6$_{1.1}$ | **79.6**$_{.20}$ | 66.2$_{1.4}$ | 49.7$_{.21}$ | 66.3$_{.17}$ | 61.76$_{.17}$ |
| CYCLEAUG | 6589 | 57.4$_{.11}$ | **35.6**$_{.34}$ | **63.9**$_{.07}$ | 29.6$_{1.0}$ | **71.2**$_{.11}$ | 1507$_{4.6}$ | 86.4$_{.82}$ | **67.0**$_{.09}$ | **61.2**$_{.26}$ | 79.3$_{.04}$ | **66.5**$_{.89}$ | **50.0**$_{.12}$ | **66.9**$_{.22}$ | **62.32**$_{.15}$ |

Table 1: **Performance comparison across thirteen benchmarks.** This table presents the results of LLaVA experiments evaluated across 13 benchmarks. For each setting, we trained 3 models and the reported results represent the mean performance along with statistics. Our method demonstrates an average improvement of 1.0% over the baseline.

## 4.3 ABLATION STUDY

In this section, we conduct an ablation study to evaluate the effectiveness of cycle-consistency sampling, the role of synthetic data as a form of regularization, and the impact of consistency judge.
**Effectiveness of cycle-consistency sampling.** To generate images from the conditional distribution

| Method | VQA$^T$ | MMMU | GQA | MMVet | SQA | MME | POPE | MMB | MMB$^{CN}$ | VQAv2 | LLaVA$^w$ | VizWiz | SEED$^I$ | Average |
|---|---|---|---|---|---|---|---|---|---|---|---|---|---|---|
| Direct sampling | 57.0 | 35.4 | **64.3** | **31.0** | 69.5 | 1496 | **87.4** | 63.9 | 56.2 | 79.0 | 66.4 | 44.0 | 63.0 | 61.01 |
| CYCLEAUG | **57.4** | **35.6** | 63.9 | 29.6 | **71.2** | **1507** | 86.4 | **67.0** | **61.2** | **79.3** | **66.5** | **50.0** | **66.9** | **62.32** |

Table 2: **Performance comparison of image generation strategies.** This table compares the performance of three image generation approaches, direct sampling and CYCLEAUG , across thirteen benchmarks. The gain shows that the quality of VQA data is crucial for training MLLMs.

defined by question-answer pairs $i \sim p(I|Q, A)$, the naive approach involves using an text-to-image generation model (stabilityai, 2024) $g_I$ where the input prompt is simply the concatenated question-answer text. However, it's difficult for $g_I$ to fully comprehend conversational context. To address this, we first convert the conversation into a descriptive image caption using a powerful language model (Dubey et al., 2024), and then generate synthetic images based on this caption. Nevertheless, this direct approach often results in a loss of diversity in the generated images, which subsequently leads to a degradation in multimodal performance. As shown in Tab.2, our proposed cycle-consistency sampling outperforms the naive approach on eight out of thirteen benchmarks and has a higher average score.

**Synthetic images as implicit regularization.** Synthetic images and their corresponding question-answer pairs, as discussed in § 3.4, are treated as regularization rather than fully aligned cross-modal data, differing from (Liu et al., 2024d). An ablation study removing real images from the training set shows that synthetic data alone yields lower performance (59.5%) compared to using real data (61.4%). Combining both achieves the best results (62.32%), highlighting the complementary role of synthetic data as a regularization tool(Tab. 3).

| Training Data | VQA$^T$ | MMMU | GQA | MMVet | SQA | MME | POPE | MMB | MMB$^{CN}$ | VQAv2 | LLaVA$^w$ | VizWiz | SEED$^I$ | Avg. |
|---|---|---|---|---|---|---|---|---|---|---|---|---|---|---|
| synthetic images | 53.5 | **36.4** | 63.8 | 29.4 | 68.4 | 1405 | 85.8 | 64.4 | 56.2 | 72.1 | **67.3** | 43.2 | 62.7 | 59.50 |
| real images | **58.2** | 35.3 | 62.0 | **31.1** | 66.8 | **1511** | 85.9 | 64.3 | 58.3 | 78.5 | 65.4 | 50.0 | 66.4 | 61.37 |
| augmented images | 57.4 | 35.6 | **63.9** | 29.6 | **71.2** | 1507 | **86.4** | **67.0** | **61.2** | **79.3** | 66.5 | **50.0** | **66.9** | **62.32** |

Table 3: **Performance comparison on different training data.** Models trained on three types of data are compared: synthetic images, real images, and augmented images (Synthetic + Real). Results are evaluated across thirteen benchmarks, with the augmented images approach achieving the highest average performance. This demonstrates the effectiveness of incorporating synthetic images as a form of regularization.

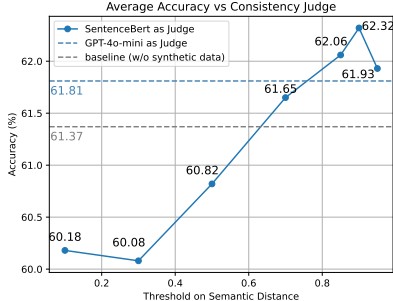

Figure 5: **Ablation on different Consistency Judges.** The blue curve depicts the relationship between average accuracy and the threshold on semantic distance. At a threshold of 0.9, the accuracy peaks at 62.32. The blue and gray dashed line represents GPT-4o-mini as a judge and baseline trained without synthetic data, respectively.The accuracy for each dataset can be found in the Appendix§ E.

**Choice of consistency judge.** Accurately determining whether a new image aligns with the original QA is critical for assessing the quality of generated data. Correspondingly, various methods are ablated in this section, including large language models–GPT-4o-mini (OpenAI, 2024)—and embedding similarity evaluation on SentenceBert (Reimers, 2019). For the first method, given the original question, original answer, and the new answer about the synthetic image, a large language model evaluates whether the two answers describe the same scene, outputting a "yes" or "no" (detailed Appendix§ D). In the second method, a threshold is applied to the normalized similarity between text embeddings of the original and new answer sentences, filtering out QA pairs with low similarity. As shown in Fig. 5, SentenceBert achieves the highest performance, with an accuracy of 62.32 at a threshold of 0.9. In comparison, GPT-4o-mini achieves an accuracy of 61.8, both outperforming

| Method | VQA$^T$ | MMMU | GQA | MMVet | SQA | MME | POPE | MMB | MMB$^{CN}$ | VQAv2 | LLaVA$^w$ | VizWiz | SEED$^I$ | Avg. |
|---|---|---|---|---|---|---|---|---|---|---|---|---|---|---|
| baseline | **58.2** | 35.3 | 62.0 | **31.1** | 66.8 | **1511** | 85.9 | 64.3 | 58.3 | 78.5 | 65.4 | 50.0 | 66.4 | 61.37 |
| Self-training CYCLEAUG | 56.6 | **35.7** | **63.9** | 30.4 | 70.7 | 1484 | 85.5 | 65.9 | 59.3 | 79.2 | **69.7** | **50.2** | 66.8 | 62.14 |
| CYCLEAUG | 57.4 | 35.6 | 63.9 | 29.6 | **71.2** | 1507 | **86.4** | **67.0** | **61.2** | **79.3** | 66.5 | 50.0 | **66.9** | **62.32** |

Table 4: **Performance comparison between baseline, self-training CYCLEAUG, and CYCLEAUG.** This table demonstrates the performance improvements achieved with CYCLEAUG and self-training CYCLEAUG across various benchmarks, highlighting the robustness of the proposed methods.

| Method | VQA$^T$ | MMMU | GQA | MMVet | SQA | MME | POPE | MMB | MMB$^{CN}$ | VQAv2 | LLaVA$^w$ | VizWiz | SEED$^I$ | Avg. |
|---|---|---|---|---|---|---|---|---|---|---|---|---|---|---|
| MGM-2B w/ cycleaug | 53.2 | 30.3 | 62.4 | 28.4 | 62.2 | 1346 | 86.1 | 60.9 | 54.4 | 78.3 | 58.4 | 47.6 | 63.5 | **58.10** |
| MGM-2B | 52.5 | 29.7 | 60.5 | 26.8 | 64.1 | 1330 | 85.8 | 59.0 | 48.5 | 77.4 | 56.7 | 47.4 | 62.4 | 56.91 |
| LLaVA-13B w/ cycleaug | 60.1 | 38.1 | 64.2 | 35.2 | 74.1 | 1517 | 87.6 | 68.9 | 64.7 | 79.9 | 73.4 | 49.5 | 67.0 | **64.55** |
| LLaVA-13B | 60.0 | 37.9 | 63.0 | 35.0 | 74.1 | 1503 | 86.6 | 68.2 | 63.5 | 79.6 | 71.0 | 53.6 | 66.8 | 64.18 |
| LLaVA-LLaMA3 w/ cycleaug | 57.4 | 37.9 | 64.7 | 35.1 | 78.9 | 1466 | 85.0 | 73.7 | 67.9 | 79.7 | 73.6 | 51.0 | 68.9 | **65.08** |
| LLaVA-LLaMA3 | 58.3 | 37.6 | 63.5 | 34.9 | 80.9 | 1468 | 86.5 | 71.3 | 66.2 | 79.5 | 71.4 | 50.0 | 69.6 | 64.85 |

Table 5: **Performance comparison across different architectures.** This table demonstrates the scalability of CYCLEAUG across various model architectures, parameter sizes, and encoder-decoder combinations. The results highlight that CYCLEAUG provides consistent improvements, with the largest gains observed in LLaVA-LLaMA3 and LLaVA-13B models.

the baseline model trained without synthetic data. Notably, performance is sensitive to the threshold value, which is vital for selecting well-aligned image and QA pairs. Without a suitable consistency judge, synthetic images can degrade model performance, even if reconstructed from detailed captions (Appendix§ D). This is because small details, such as objects and colors, are not always faithfully reconstructed, introducing noise into the dataset. A robust consistency judge is essential to bridge the alignment gap caused by limitations in both the diffusion and caption models.

**Self-training performance.** While CYCLEAUG outperforms the baseline, it is important to determine whether this performance gain originates from CYCLEAUG itself or from distilling knowledge from more advanced MLLMs, such as LLaVA-NEXT. To test this, LLaVA-v1.5-7b is utilized as both the image captioner and judge, placing CYCLEAUG in a self-training setting. As shown in Table ???4, the results highlight the robustness and adaptability of CYCLEAUG under self-training.

**Scalability to different architectures.** In this section, we validate CYCLEAUG's performance across three architectures: LLaVA-LLaMA-3, MGM-2B, and LLaVA-13B. This evaluation demonstrates CYCLEAUG's scalability under varying parameter counts, visual encoders, and language models. As shown in Tab. 5, CYCLEAUG consistently improves performance across all architectures, particularly excelling in larger models and diverse setups, highlighting its adaptability.

# 5 CONCLUSION

In this paper, we have conducted a study on generating synthetic images to enhance the performance of multimodal LLMs. We introduced a novel cross-modal data augmentation approach leveraging cycle-consistency sampling, which plays a vital role in generating high-quality and well-aligned cross-modality data and acts as an implicit regularization mechanism during training. Our extensive experiments demonstrate that this approach improves model performance across multiple visual question answering benchmarks, achieving an average performance gain of 1% without relying on additional real-world data. We hope this work inspires future research on the creation of high-quality synthetic data for multimodal LLMs.

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

# A FLOPs AND TIME-COST

In this section, we measure the FLOPs and time cost of CYCLEAUG to evaluate its computational efficiency. As shown in Table 6, the image generation step (Stable Diffusion 3) accounts for the highest computational cost, requiring 133.9 GPU-hours due to its high FLOPs (87.9T) and latency (4.1s/sample). Image captioning and VQA (LLaVA-Llama-3) strike a balance, processing 273,144 samples in 91.0 GPU-hours, with moderate FLOPs (8.3T) and latency (1.2s/sample). The consistency judgment step (Sbert-Model) is the most efficient, contributing only 3.7 GPU-hours.This analysis highlights that image generation is the main bottleneck, while captioning and consistency judgment remain computationally efficient. Optimizing the generation step could significantly reduce overall cost, improving scalability for larger datasets.

| Model | FLOPS (T) | Latency (s/sample) | # Images / # QA | Overall Time (h * GPU) |
|---|---|---|---|---|
| SD *for image generation* | 87.9 | 4.1 | 117,576 | 133.9 |
| LLaVA-llama-3 *for image captioning & VQA* | 8.3 | 1.2 | 273,144 | 91.0 |
| Sbert-Model *for consistency judge* | 0.2 | 0.0 | 273,144 | 3.7 |

Table 6: **Performance of different models for data generation, captioning, and consistency judgment.** The table shows the FLOPS, latency, number of processed images and QA pairs, total time in seconds, and GPU-hours for each model.

# B OTHER AUGMENTATIONS

Addition to direct sampling and CYCLEAUG, we introduce a third approach, referred to as "composed transformation," where generated images are further augmented using a combination of random crop and resize, rotation, flip, and color jitter. This method enhances image diversity in appearance. However, such augmentation is likely to harm the alignment between image and QA like changing the spatial relationship after image rotating. Therefore, it don't have remarkably positive improvment on VQA task(Tab. 7).

# C PROOF OF LEMMAS

**Lemma 1.** We start from formulating our proposal distribution for cycle-consistency sampling. We choose $q$ from existing questions and retrieve the corresponding images $i$, which could be more than one, for each $q$, effectively sampling from $p(I|q)$. We then use existing MLLMs to generate captions $c \sim p(C)$ for the original images $i$ with prompt $x$, which we interpret as sampling from $p(C|I,x)$. Since the prompt $x$ is consistently set to "Describe the image in detail for reconstruction," we simplify the notation $p(C|I,x)$ to $p(C|I)$ in the subsequent discussion. After that, we use diffusion models with textual inputs $C$ to generate synthetic images $I'$, sampling from $p(I'|C)$. We then state two initial assumptions, which are the captioning process is independent of the selected question $q$ with image $I$ as in Eq.8 and the image generation process depends only on $C$, being independent of $I$ and $q$ as in Eq.9.

$$p(C|I,q) = p(C|I) \tag{8}$$
$$p(I'|C,I,q) = p(I'|C) \tag{9}$$

Given these assumptions, our proposal distribution for image sampling can be written as Eq.10:

$$p(I'|q) = \int \int p(I'|C)\, p(C|I)\, p(I|q)\, dI\, dC = \int K(I'|I)\, p(I|q)\, dI, \tag{10}$$

where $K(I'|I)$ is the transformation kernel defined as:

$$K(I'|I) = \int p(I'|C)\, p(C|I)\, dC. \tag{11}$$

We assume that the caption $C$ retains most of the information about $I$ and the generation process $p(I'|C)$ reproduces that information accurately in $I'$. In other words, we define $C$ as a sufficient

| Method | VQA$^T$ | MMMU | GQA | MMVet | SQA | MME | POPE | MMB | MMB$^{CN}$ | VQAv2 | LLaVA$^w$ | VizWiz | SEED$^I$ | Avg. |
|---|---|---|---|---|---|---|---|---|---|---|---|---|---|---|
| Composed Transformation | 57.0 | 34.3 | 62.9 | 28.8 | 70.8 | 1501 | 85.6 | 66.0 | 60.2 | 79.1 | 64.5 | 48.5 | 66.1 | 61.44 |
| Direct sampling | 57.0 | 35.4 | **64.3** | **31.0** | 69.5 | 1496 | **87.4** | 63.9 | 56.2 | 79.0 | 66.4 | 44.0 | 63.0 | 61.01 |
| CYCLEAUG | **57.4** | **35.6** | 63.9 | 29.6 | **71.2** | **1507** | 86.4 | **67.0** | **61.2** | **79.3** | 66.5 | **50.0** | 66.9 | **62.32** |

Table 7: **Performance comparison of image generation strategies.** This table compares the performance of three image generation approaches, composed transformation, direct sampling and CYCLEAUG , across thirteen benchmarks. The gain shows that the quality of VQA data is crucial for training MLLMs.

statistic of $I$ for $I'$ when conditioned on $q$. As a result, $K(I'|I)$ is sharply peaked around $I' = I$, which implies that for each original image $I$, the synthetic image $I'$ generated through the captions is very close to $I$, leading to:

$$K(I'|I) \approx \delta(I' - I) \tag{12}$$

To prove $p(I'|q) \approx p(I|q)$, without loss of generality, $\forall i_0 \in I'$, where $i_0$ is an arbitrary synthetic image, we have by plugging eq. 12 back eq. 10:

$$p(I' = i_0|q) \approx \int \delta(i_0 - I)\, p(I|q)\, dI \tag{13}$$

$$= p(I = i_0|q) \tag{14}$$

Therefore, $\forall i_0 \in I'$ we have $p(I' = i_0|q) \approx p(I = i_0|q)$ and then $p(I'|q) \approx p(I|q)$.

Our proposal distribution for sampling $p(I'|q)$ can be approximated to $p(I|q)$. We then define the image pair $(i, i') \sim p(I|q)$ to be cycle-consistent with respect to the question-answer pair $(q, a)$ if and only if

$$a = a' := f_\psi(i', q), \tag{15}$$

where $f_\psi(\cdot)$ is a pretrained MLLM.

Subsequently, we illustrate the equivalence of our proposed cycle-consistency sampling and sampling from the conditional distribution $p(I|Q, A)$. We start by expressing our target posterior distribution using Bayes' Theorem in Eq.16.

$$p(I|q, a) = \frac{p(a|I, q)p(I|q)}{p(a|q)} \tag{16}$$

As we are able to evaluate drawn samples $i'$ in the semantic space; in other words, we can derive $a$ for any $i$ and corresponding $q$, as $a = f_\psi(i, q)$, we can then rewrite the likelihood as in Eq.17, in which $\delta(\cdot)$ refers to the Dirac Delta function.

$$p(a|I = i, q) = \delta(a - f_\psi(i, q)) \tag{17}$$

Then, we formulate the evidence $p(a|q)$ in Eq.18.

$$p(a|q) = \int p(a|I = i, q)p(I = i|q)di = \int \delta(a - f_\psi(i, q))p(i|q)di = \int_{\{i|f_\psi(i,q)=a\}} p(i|q)\, di \tag{18}$$

Therefore, we can rewrite the posterior distribution into Eq.19.

$$p(I = i \mid q, a) = \frac{p(a \mid i, q)p(i|q)}{p(a \mid q)} = \frac{\delta(a - f_\psi(i, q))p(i|q)}{\int_{\{i|f_\psi(i,q)=a\}} p(i|q)\, di} \tag{19}$$

**Lemma 2.** We then demonstrate that synthetic images from the proposed conditional distribution effectively act as an implicit regularization for training MLLMs.

During instruction tuning, multimodal large language models (MLLMs) learn to identify relevant visual elements in images that correspond to the provided questions, which represents the mutual information between visual and textual modalities. Given pairs $(i, q, a)$ sampled from the joint distribution $p(I, Q, A)$, the training objective can be expressed as:

$$\mathcal{L} = \mathbb{E}_{(i,q,a)\sim p(I,Q,A)} \left[ \ell(f_\theta(i, q), a) \right] \tag{20}$$

, in which $\ell(\cdot)$ is a general form of loss objectives used to reduce the distance between $f_\theta(i, q)$ and $a$ during optimization.

We propose to use synthetic images sampled from the conditional distribution $p(I|Q, A)$ with cycle consistency to augment the training sets, effectively adding an implicit regularization for the objective. After adding samples from the conditional distribution, our training objective alters to:

$$\mathcal{L}' = \mathbb{E}_{(q,a)\sim p(Q,A)} \left[ \mathbb{E}_{i'\sim p(I|q,a)} \left[ \ell(f_\theta(i', q), a) \right] \right] \tag{21}$$

We can rewrite Eq.21 into the original objective and extra terms by adding and subtracting $\mathbb{E}_{(i,q,a)\sim p(I,Q,A)} \left[ \ell(f_\theta(i, q), a) \right]$:

$$\begin{aligned}
\mathcal{L}' &= \mathbb{E}_{(i,q,a)\sim p(I,Q,A)} \left[ \ell(f_\theta(i, q), a) \right] \\
&+ \left( \mathbb{E}_{(q,a)\sim p(Q,A)} \left[ \mathbb{E}_{i'\sim p(I|q,a)} \left[ \ell(f_\theta(i', q), a) \right] \right] - \mathbb{E}_{(i,q,a)\sim p(I,Q,A)} \left[ \ell(f_\theta(i, q), a) \right] \right)
\end{aligned}$$

After further simplication, $L'$ can be rewritten as:

$$\mathcal{L}' = \mathcal{L} + \mathbb{E}_{(q,a)\sim p(Q,A)} \left[ \mathbb{E}_{i'\sim p(I|q,a)} \left[ \ell(f_\theta(i', q), a) \right] - \mathbb{E}_{i\sim p(I|q,a)} \left[ \ell(f_\theta(i, q), a) \right] \right] \tag{22}$$

We then define a semantic distance $d$ between $f(i', q)$ and $f(i, q)$, which is anchored by the ground-truth answer $a$, as follows:

$$d(i, i'|q, a) = |\ell(f_\theta(i', q), a) - \ell(f_\theta(i, q), a)| \tag{23}$$

By definition, the regularization term $R$ is rewritten as in Eq.24, which leads to the consistency regularization. Here, we assume that $\mathbb{E}_{i'\sim p(I|q,a)} \left[ \ell(f_\theta(i', q), a) \right]$ is greater than $\mathbb{E}_{i\sim p(I|q,a)} \left[ \ell(f_\theta(i, q), a) \right]$, as synthetic images $\{i'\}$ are considered to be noisier than real images $\{i\}$ for $(q, a)$.

$$R = \mathbb{E}_{(q,a)\sim p(Q,A)} \left[ \mathbb{E}_{i,i'\sim p(I|q,a)} \left[ d(i, i'|q, a) \right] \right] \tag{24}$$

Therefore, incorporating synthetic images generated from the conditional distribution $p(I|Q, A)$ effectively introduce an implicit consistency regularization, which encourages MLLMs to generate consistent answers for the image pairs.

# D    QUALITATIVE EXAMPLES

In this section, we provide more examples about qualitative comparison between CYCLEAUG and Direct Sampling. As shown in Fig. 6, we present a visual comparison between images generated using cycle-consistency sampling and direct sampling, with reference to a set of anchor images. The top row shows the anchor images, which serve as the initial inputs for comparison. The middle row displays the results from cycle-consistency sampling, which closely align with the content and context of the anchor images, demonstrating semantic consistency and relevance. In contrast, the bottom row shows images generated through direct sampling, which exhibit more randomness and less adherence to the original content of the anchor images. This comparison visually highlights the effectiveness of cycle-consistency sampling in maintaining the integrity of the generated images in relation to the reference anchor images.

We also provide the prompts we use for image generation and consistency judge(Fig. 7).

Anchor Image

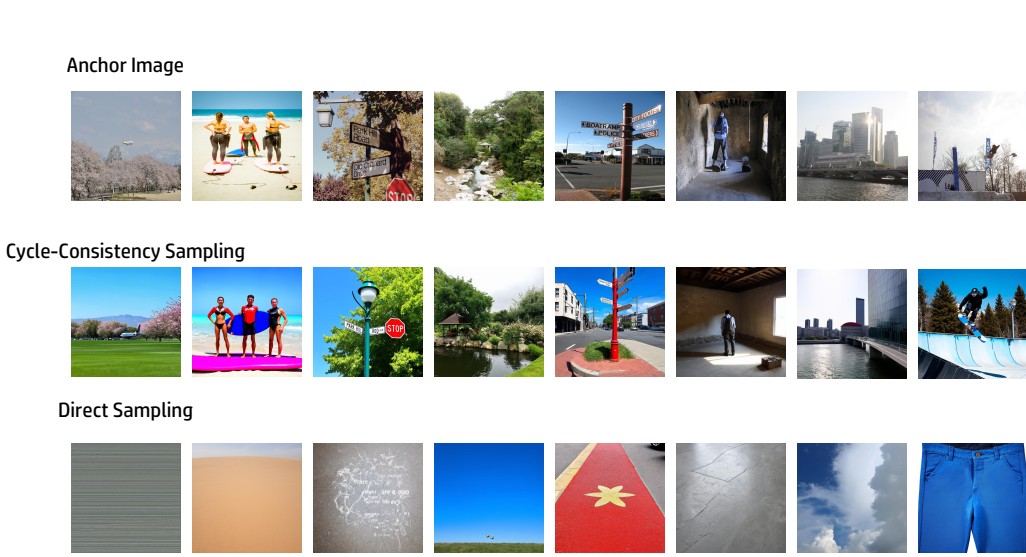

Cycle-Consistency Sampling

Direct Sampling

Figure 6: **Comparison of image generation methods.** The top row shows the anchor images used as the base reference. The middle row presents the images generated through CYCLEAUG, which maintains semantic alignment with the anchor images. The bottom row shows the results of direct sampling, which lacks the same level of consistency and often diverges from the content of the anchor images. CYCLEAUG demonstrates improved relevance and coherence in generating images that reflect the context of the original anchors.

> Prompt for Image Captioning:
>
> 1. Describe the image in details for image reconstruction.
> 2. Provide a concise and precise description of the image so that others can recreate it from the description.
> 3. Use brief and accurate language to fully describe the image, allowing people to recreate it based on the description.
> 4. Give a short and accurate description of the image, enabling others to recreate the image from it.
> 5. Describe the image succinctly and clearly, allowing people to reconstruct it using the description.
>
> Prompt for Consistency judge with GPT-4o-mini as judge:
>
> You are an assistant with strong logical abilities. You will be provided with a question and a pair of answers. Your task is to determine whether the two answers are consistent or if they describe the same image with respect to the provided question

Figure 7: Prompts for image generation and consistency judge

In the background, a yellow building with a green roof adds a splash of color to the scene. The building's vibrant hues contrast with the more muted tones of the surrounding structures, drawing attention to itself. The street itself is lined with power lines, a common sight in many urban areas. They crisscross above the street, creating a network that is both functional and visually interesting. The sky above is a clear blue, suggesting a bright and sunny day. The absence of clouds indicates good weather, which is often associated with increased activity on the streets. Overall, the image paints a picture of a bustling city street in Bogota, with its mix of commercial signs, colorful buildings, and clear blue sky.

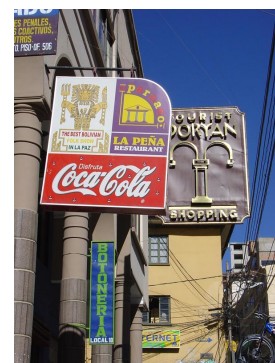

Figure 8: Prompts for top row in Fig. 2

In the tranquil setting of a zoo enclosure, two polar bears are captured in a moment of rest. The bear on the left, with its back to us, is lying on a rock formation, its body relaxed and at ease. Its fur, a mix of white and light brown, blends harmoniously with the natural surroundings. On the right, another bear is seen lying on a bed of green leaves. Its head is gently resting on the leaves, suggesting a sense of comfort and contentment. The leaves, a vibrant green, provide a stark contrast to the bear's white and brown fur.The enclosure itself is designed to mimic the bears' natural habitat. A large glass window forms the backdrop, allowing visitors to observe the bears from a safe distance. The window reflects the lush greenery outside, further enhancing the natural ambiance of the enclosure.The image is a beautiful snapshot of life in the zoo, capturing the serene moments of these majestic creatures in their man-made habitat.

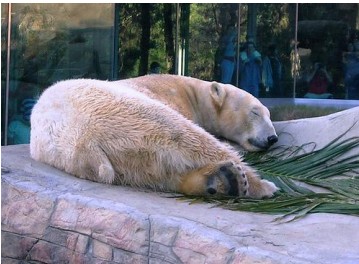

Figure 9: Prompts for bottom row in Fig. 2

# E    ADDITIONAL EXPERIMENTAL RESULTS

In this section, we provides the quantitive result of the impact of consistency judge(see § 4.3)

We provide the caption used for generating image shown in Fig. 2 through CYCLEAUG.

| Consistency Judge | VQA$^T$ | MMMU | GQA | MMVet | SQA | MME | POPE | MMB | MMB$^{CN}$ | VQAv2 | LLaVA$^w$ | VizWiz | SEED$^I$ | Avg. |
|---|---|---|---|---|---|---|---|---|---|---|---|---|---|---|
| w/o synthetic data | 58.2 | 35.3 | 62.0 | **31.1** | 66.8 | **1510.7** | 85.9 | 64.3 | 58.3 | 78.5 | 65.4 | 50.0 | 66.4 | 61.37 |
| 0.1 | **58.3** | 34.7 | 63.0 | 29.4 | 70.9 | 1514 | 85.9 | 66.3 | 57.6 | 59.9 | 64.2 | 49.0 | 66.4 | 60.10 |
| 0.3 | 57.5 | 34.7 | 63.2 | **31.3** | 71.0 | 1458 | 86.4 | 65.6 | 57.3 | 59.9 | 65.2 | 49.9 | 66.0 | 60.07 |
| 0.5 | **58.3** | 34.6 | 63.2 | 30.1 | 69.0 | 1495 | 86.2 | 65.3 | 58.3 | 72.9 | 64.2 | 50.1 | 64.0 | 60.83 |
| 0.7 | 58.1 | 34.7 | 63.2 | 29.6 | 70.7 | 1482 | 86.6 | 66.0 | 59.8 | 76.3 | 66.2 | **50.5** | 66.2 | 61.68 |
| 0.85 | 57.6 | **36.6** | 63.0 | 29.1 | **71.9** | **1521** | 86.6 | 66.2 | **61.2** | 79.3 | 63.3 | 50.2 | 65.8 | 62.07 |
| 0.9 | 57.4 | 35.6 | **63.9** | 29.6 | 71.2 | 1507 | **86.4** | **67.0** | **61.2** | 79.3 | 66.5 | 50.0 | **66.9** | **62.32** |
| 0.95 | 58.0 | 33.7 | 63.5 | 30.3 | 70.6 | 1507 | 86.5 | 65.9 | 58.5 | **79.3** | **67.6** | 49.8 | 65.8 | 61.91 |
| GPT-4o-mini | 58.2 | 35.2 | 62.7 | 31.0 | 70.4 | 1511 | 85.6 | 66.4 | 58.8 | **79.3** | 65.4 | 50.01 | 66.3 | 61.9 |

Table 8: **Performance comparison on different Consistency Judges**. This table compares the performance of models trained on different consistency judge including threshold on different semantic distance as well as GPT-4o-mini-based judgement.

