# OpenReview forum: "CycleAug: Cycle-Consistent Visual Augmentation for Large Multimodal Models"
_ICLR.cc/2025/Conference — Submitted to ICLR 2025_

### Official Review · Reviewer_pwkd · 2024-10-16

**Soundness:** 2
**Presentation:** 3
**Contribution:** 2
**Rating:** 3
**Confidence:** 3

**Summary:**

This paper proposes a new visual augmentation strategy for improving the MLLM's generalization ability. Specifically, it introduces a cycle-consistency sampling process, which utilizes a three-step process to collect new images. In the first step, it generates images based on the caption/question; then in the second step, it measures the consistency between the new image and the original caption/question. At last, the positive samples will be added to the datasets for training. The authors also exploit implicit Bayesian to interpret the generation process. Experiments are conducted on thirteen benchmarks.

**Strengths:**

1. The paper is easy to follow.

2. The motivation is reasonable.

**Weaknesses:**

1. The novelty is weak. The consistency sampling is directly inspired by the existing cross-modal consistency alignment framework. Although the idea is straightforward, there is no special or novel technical design, weakening the novelty of the proposed method.

2. The implicit Bayesian interpretation is good, however, it is not the main contribution of this paper in my opinion. The authors just did a straightforward thing to generate synthetic data to boost the MLLMs, which is reasonable and easy to understand. Utilizing technique-irrelevant formulations and proof is not necessary.

3. The authors only consider the visual augmentation via the loop of text-image-text. What about generating synthetic texts via the loop of image-text-image to provide new aspects of the same image?

4. Experiments are unconvincing. Firstly, the authors only conduct the experiments on a single MLLM model. At least 3-4 MLLM models should be performed to evaluate the proposed method fairly. Secondly, as shown in Table 1, the proposed method fails to achieve the SOTA on FIVE important benchmarks. Thirdly, the authors should provide a detailed complexity and efficiency analysis of the cycle-consistency visual augmentation process.

5. The authors should provide discussions about the failure cases and their limitations. Not all generated images are positive to the MLLM training. A more in-depth analysis is also required to be provided.

**Questions:**

1. The novelty is weak. The consistency sampling is directly inspired by the existing cross-modal consistency alignment framework. Although the idea is straightforward, there is no special or novel technical design, weakening the novelty of the proposed method.

2. The implicit Bayesian interpretation is good, however, it is not the main contribution of this paper in my opinion. The authors just did a straightforward thing to generate synthetic data to boost the MLLMs, which is reasonable and easy to understand. Utilizing technique-irrelevant formulations and proof is not necessary.

3. The authors only consider the visual augmentation via the loop of text-image-text. What about generating synthetic texts via the loop of image-text-image to provide new aspects of the same image?

4. Experiments are unconvincing. Firstly, the authors only conduct the experiments on a single MLLM model. At least 3-4 MLLM models should be performed to evaluate the proposed method fairly. Secondly, as shown in Table 1, the proposed method fails to achieve the SOTA on FIVE important benchmarks. Thirdly, the authors should provide a detailed complexity and efficiency analysis of the cycle-consistency visual augmentation process.

5. The authors should provide discussions about the failure cases and their limitations. Not all generated images are positive to the MLLM training. A more in-depth analysis is also required to be provided.

---

> ### Author Response · Authors · 2024-11-30
> **Part 1: Reponse to Weakness 1-3 and Question 1-3**
>
> We appreciate the reviewer's constructive suggestions and apologize for the delayed response, as we have been preparing experiments to thoroughly address your concerns. In addition, we have revised our manuscript based on your suggestions.
>
> > W1: The novelty is weak. The consistency sampling is directly inspired by the existing cross-modal consistency alignment framework. Although the idea is straightforward, there is no special or novel technical design, weakening the novelty of the proposed method.
>
> We thank the reviewer for the opportunity to clarify the novelty and contributions of our work. However, we respectfully disagree with the assertion that our method is directly inspired by existing cross-modal consistency alignment frameworks.
>
> We would greatly appreciate it if the reviewer could elaborate further on these frameworks for better context. Our work does not aim to develop a new multimodal alignment approach. To clarify, our work represents the first attempt to augment the training of large multimodal models using cycle-consistency. To the best of our knowledge, no prior research has explored this approach for training large multimodal models.
>
> > W2: Utilizing technique-irrelevant formulations and proof is not necessary.
>
> Thank you for highlighting the distinction between our mathematical formulation and the implementation techniques. While we acknowledge that the mathematical formulation may not directly align with every aspect of the implementation, **it serves as a precise and clear expression of our underlying motivation**.
>
> Inspired by analogical reasoning, our method aims to encourage feature-space consistency for semantically similar yet visually distinct image pairs. This concept is captured in Lemma 2, providing a theoretical view for our approach.
>
> To ensure that the paper remains focused and accessible, we have included the full proof in the appendix while presenting the key conclusions in the main text. This strategy maintains the clarity and flow of the article while offering a deeper theoretical perspective for interested readers. Furthermore, the experimental results validate the practical effectiveness of this regularization, demonstrating its importance in enhancing the performance of the model.
>
> > W3: generating synthetic texts via the loop of image-text-image to provide new aspects of the same image
>
> Thank you for the thoughtful suggestion regarding generating synthetic texts via the loop of image-text-image to provide new perspectives on the same image. While our current work focuses on visual augmentation through the text-image-text loop, we agree that exploring the reverse loop (image-text-image) is a promising direction for generating diverse textual data. This idea could enrich multimodal learning by introducing additional dimensions of understanding.
>
> However, we respectfully note that this direction falls beyond the scope of our current work, which is specifically centered on visual augmentation and its impact on multimodal learning models. We kindly request that aspects unrelated to the core contributions of the paper not be considered as weaknesses, as they do not detract from the novelty or rigor of our proposed method. We look forward to exploring this idea further in future research.

---

> ### Author Response · Authors · 2024-11-30
> **Part 2: Response to Weakness 4 and Question 4**
>
> > W4:  At least 3-4 MLLM models should be performed to evaluate the proposed method fairly. authors should provide a detailed complexity and efficiency analysis of the cycle-consistency visual augmentation process.
>
> To address the reviewer’s concerns, we have conducted additional experiments across multiple MLLMs, including LLaVA-LLaMA3, LLaVA-v1.5-13B, and MGM-2B:
> | Model | $VQA^T$ | MMMU | GQA | MM-VET | SQA | MME | POPE | MMB | $MMB^{CN}$ | $VQA_{v2}$ | $LLaVA^w$ | VizWiz | $SEED_I$ | Average |
> |----------------------------|-----------|----------|----------|------------|------------|---------|----------|-----------|---------|-----------|------------|------------|----------|----------|
> | MGM-2B (w/ cycleaug)| 53.2 | 30.3 | 62.4 | 28.4 | 62.2 | 1346 | 86.1 | 60.9 | 54.4 | 78.3 | 58.4 | 50.0 | 63.5 | **58.10** |
> | MGM-2B  | 52.5 | 29.7 | 60.5 | 26.8 | 64.1 | 1330 | 85.8 | 59.0 | 48.5 | 77.4 | 56.7 | 50.0 | 62.4 | 56.91 |
> | LLaVA-13B (w/ cycleaug)| 60.1 | 38.1 | 64.2 | 35.2 | 74.1 | 1517 | 87.6 | 68.9 | 64.7 | 79.9 | 73.4 | 50.0 | 67.0 | **64.55** |
> | LLaVA-13 | 60.0 | 37.9 | 63.0 | 35.0 | 74.1 | 1503 | 86.6 | 68.2 | 63.5 | 79.6 | 71.0 | 53.6 | 66.8 | 64.85 |
> | LLaVA-LLaMA3 (w/ cycleaug)| 57.4 | 37.9 | 64.7 | 35.1 | 78.9 | 1466 | 85.0 | 73.7 | 67.9 | 79.7 | 73.6 | 50.0 | 68.9 | **65.08** |
> | LLaVA-LLaMA3 | 58.3 | 37.6 | 63.5 | 34.9 | 80.9 | 1468 | 86.5 | 71.3 | 66.2 | 79.5 | 71.4 | 50.0 | 69.6 | 64.85 |
>
> **Performance Gains**: Our method consistently outperforms baselines, with notable improvements on GQA, MMB, and MM-Vet benchmarks. Average performance gains are +1.19%, +0.34%, and +0.23% for MGM-2B, LLaVA-13B, and LLaVA-LLaMA3, respectively.
>
> We acknowledge the importance of providing a detailed complexity and efficiency analysis for the cycle-consistency visual augmentation process. This analysis demonstrates that our method is efficient relative to the scale of augmentation and its benefits:
>
> **Cost Analysis**:
>
> The computational cost is estimated based on the FLOPs and latency of each component, averaged across 200 random samples, and scaled to the total dataset size. The measurements are as follows:
>
> | Model                          | FLOPS (T)  | Latency (s/sample) | # Images/QA | Overall Time (s) | Overall Time (h·GPU) |
> |--------------------------------|------------|---------------------|-------------|------------------|-----------------------|
> | SD3 (Image Generation)          | 87.9       | 4.1                 | 117,576     | 482,062          | 133.9                 |
> | VAR-d-30[1] (Image Generation)   | 2.3        | 0.7                 | 117,576     | 82,303           | 22.9                   |
> | LLaVA-LLaMA-3 (Captioning & VQA) | 8.3       | 1.2                 | 273,144     | 327,773          | 91.0                  |
> | SBERT Model (Consistency Judge) | 0.2        | 0.0                 | 273,144     | 13,250           | 3.7                   |
>
> [1] Visual autoregressive modeling: Scalable image generation via next-scale prediction. 2024.
>
> **FLOPs and time-cost distribution**:
> We measure the FLOPs and time cost of CycleAug to evaluate its computational efficiency. As shown in Table, the image generation step (Stable Diffusion 3) accounts for the highest computational cost, requiring 133.9 GPU-hours due to its high FLOPs (87.9T) and latency (4.1s/sample). Image captioning and VQA (LLaVA-Llama-3) strike a balance, 91.0 GPU-hours, with moderate FLOPs (8.3T) and latency (1.2s/sample). The consistency judgment step (Sbert-Model) is the most efficient, contributing only 3.7 GPU-hours.This analysis highlights that image generation is the main bottleneck, while captioning and consistency judgment remain computationally efficient. Optimizing the generation step could significantly reduce overall cost, improving scalability for larger datasets.
>
> **Large Room for FLOPs reduction**:
> Although the FLOPs and time costs are significant for Stable Diffusion 3, we also measured the performance of VAR, an autoregressive-based text-to-image generation method with comparable quality to diffusion-based methods (FID: 1.8 vs. 1.56). VAR requires only 22.9 GPU hours to generate 117k images, making it both efficient and scalable to larger datasets.
>
> Since our current focus is on demonstrating the proof of concept, we have not emphasized efficiency. However, **the performance of VAR highlights the potential for our pipeline to achieve high efficiency**, especially when integrated with more computationally efficient models.
>
> **Conclusion**:
> CycleAug's computational cost is significant, primarily due to the image generation step, but the filtering process ensures high-quality data critical for performance improvements. Efficient alternatives like VAR, with comparable quality and lower costs, offer potential for optimization, making CycleAug more scalable for larger datasets.

---

> ### Author Response · Authors · 2024-11-30
> **Part 3: Weakness 5 and Question 5**
>
> > W5: The authors should provide discussions about the failure cases and their limitations.
>
> We appreciate the opportunity to discuss the limitations.
>
> (1) Text-rich VQA: The current version might not handle well with text-rich VQA. This stems from the common limitations of the Stable Diffusion models in generating characters, which hinders its performance on text-focused QA tasks like TextVQA. However, we believe this limitation is not inherent to our pipeline but rather tied to the current capabilities of image generation models. As these models continue to improve in text rendering (such as [ideogram](https://ideogram.ai/) and [Text-Diffuser2](https://arxiv.org/pdf/2311.16465)), we expect this issue to be resolved.
>
> (2) Misaligned Samples: This occurs when the cycle-consistency sampling process does not completely filter out all misaligned images. While our consistency judge effectively identifies and removes most misaligned samples with respect to the question-answer pair, occasional errors arise from imperfections in the underlying vision-language model and image generation model. Nonetheless, our carefully designed consistency judge successfully identifies the majority of valid samples in an automatic and efficent way, improving the model's performance on VQA tasks.
>
> Through analysis, we observe that the negative impact of erroneous images is most pronounced in datasets requiring precise fine-grained visual-text alignment, such as text-rich scenarios. Conversely, the impact is minimal on benchmarks with a broader focus, such as GQA, MMB, VQAv2, where diversity in visual content is a more critical factor.

---

> ### Author Response · Authors · 2024-12-02
>
> Dear Reviewer,
>
> If our response does not address your remaining concerns, please let us know, and we will address them promptly before the rebuttal period concludes. In addition, we have revised our manuscript based on your suggestions.
>
> Thank you!

---

> > ### Comment · Reviewer_pwkd · 2024-12-02
> > **Reply to Authors**
> >
> > Thanks for your responses! Some of my concerns are addressed. However, the cycle consistency is not new, and the authors fail to discuss it with existing cross-modal cycle consistency works that share similar techniques. Therefore, I can not raise my score at this time.

---

### Official Review · Reviewer_aJx2 · 2024-10-29

**Soundness:** 3
**Presentation:** 3
**Contribution:** 2
**Rating:** 5
**Confidence:** 5

**Summary:**

This paper propose CYCLEAUG, a data augmentation framework for MLLM training. CYCLEAUG first generate new images based on the image caption of the source image-text data, then use an MLLM to check the consistency of the generated images with the original image, and filter the inconsistent ones. The authors use the filtered augmented data to train a LLaVA1.5 and show the effectiveness of the generated data in MLLM training.

**Strengths:**

Data augmentation/generation for MLLM training is an important problem. The experiment results prove the effectiveness of the augmented data output by the proposed CYCLEAUG.

**Weaknesses:**

1. The computational cost of the whole pipeline might be too heavy.
2. The main experiment is not convincing. The authors use LLama3-LLaVA-NeXT-8B for image caption (line 388), and LLaVA-NeXT for for consistency judge (according to the citation in line 308). However, the augmented data are used to train a LLaVA-1.5-7B, which uses a way worse base LLM (Vicuna-7B) and a weaker structure than LLaVA-NeXT. The potential assumption is that there exists a better MLLM of similar scale compared to the MLLM we are going to train. If so, why can't we use the better MLLM directly?

**Questions:**

1. What is the approximate computational cost for the data augmentation by CYCLEAUG in the main experiment?
2. Since the authors use better MLLM in the CYCLEAUG framework, is the performance gain from the CYCLEAUG framework itself, or from distilling knowledge from the better MLLM (LLaVA-NeXT) to the MLLM we are training (LLaVA-1.5-7B)?

---

> ### Author Response · Authors · 2024-11-30
> **Part 1: Response to Weakness 1 and Question 1**
>
> We appreciate the reviewer’s constructive suggestions and apologize for the delayed response, as we have been preparing experiments to thoroughly address your concerns. In addition, we have revised our manuscript based on your suggestions.
>
> > W1 & Q1: approximate computational cost for the data augmentation by CYCLEAUG in the main experiment
>
> We appreciate the reviewer’s suggestion to report the computational cost and filtering proportions, which we have now calculated. Below, we detail the FLOPs, latency, and overall time for each stage of our process, along with the proportion of images filtered out by cycle-consistency sampling:
>
> **Cost Analysis**:
> The computational cost is estimated based on the FLOPs and latency of each component, averaged across 200 random samples, and scaled to the total dataset size. The measurements are as follows:
> | Model                          | FLOPS (T)  | Latency (s/sample) | # Images/QA | Overall Time (s) | Overall Time (h·GPU) |
> |--------------------------------|------------|---------------------|-------------|------------------|-----------------------|
> | SD3 (Image Generation)          | 87.9       | 4.1                 | 117,576     | 482,062          | 133.9                 |
> | VAR-d-30[1] (Image Generation)   | 2.3        | 0.7                 | 117,576     | 82,303           | 22.9                   |
> | LLaVA-LLaMA-3 (Captioning & VQA) | 8.3       | 1.2                 | 273,144     | 327,773          | 91.0                  |
> | SBERT Model (Consistency Judge) | 0.2        | 0.0                 | 273,144     | 13,250           | 3.7                   |
>
> [1] Visual autoregressive modeling:Scalable image generation via next-scale prediction. 2024.
>
> **Filtering Proportion**:
> Out of the 273,144 images initially generated, **117,576 images (43%)** were retained after applying cycle-consistency sampling. This step effectively eliminates 57% of the images, ensuring the final dataset is both diverse and of high quality.
>
> **FLOPs and time-cost distribution**:
> We measure the FLOPs and time cost of CycleAug to evaluate its computational efficiency. As shown in Table, the image generation step (Stable Diffusion 3) accounts for the highest computational cost, requiring 133.9 GPU-hours due to its high FLOPs (87.9T) and latency (4.1s/sample). Image captioning and VQA (LLaVA-Llama-3) strike a balance, processing 273,144 samples in 91.0 GPU-hours, with moderate FLOPs (8.3T) and latency (1.2s/sample). The consistency judgment step (Sbert-Model) is the most efficient, contributing only 3.7 GPU-hours.This analysis highlights that image generation is the main bottleneck, while captioning and consistency judgment remain computationally efficient. Optimizing the generation step could significantly reduce overall cost, improving scalability for larger datasets.
>
> **Large Room for FLOPs reduction**:
> Although the FLOPs and time costs are significant for Stable Diffusion 3, we also measured the performance of VAR, an autoregressive-based text-to-image generation method with comparable quality to diffusion-based methods (FID: 1.8 vs. 1.56). VAR requires only 22.9 GPU hours to generate 117k images, making it both efficient and scalable to larger datasets.
>
> Since our current focus is on demonstrating the proof of concept, we have not emphasized efficiency. However, **the performance of VAR highlights the potential for our pipeline to achieve high efficiency**, especially when integrated with more computationally efficient models.
>
> **Conclusion**:
> CycleAug's computational cost is significant, primarily due to the image generation step, but the filtering process ensures high-quality data critical for performance improvements. Efficient alternatives like VAR, with comparable quality and lower costs, offer potential for optimization, making CycleAug more scalable for larger datasets.

---

> ### Author Response · Authors · 2024-11-30
> **Part2: Response to Weakness 2 and Question 2**
>
> > W2 & Q2: Since the authors use better MLLM in the CYCLEAUG framework, is the performance gain from the CYCLEAUG framework itself, or from distilling knowledge from the better MLLM (LLaVA-NeXT) to the MLLM we are training (LLaVA-1.5-7B)
>
> Thank you for pointing out this important consideration. We acknowledge the concern about the difference in models used for captioning and consistency judgment compared to the target model being trained.
>
> In our current experimental setup, we use stronger models (e.g., LLaVA-NeXT) for image captioning and consistency judgment to ensure high-quality augmented data. However, we want to emphasize that our approach is not limited to this setting. It can also operate effectively in a self-training scenario where the captioner and judge are the same as the target model, eliminating the need for a better external model.
>
> **To address this, we conducted additional experiments using LLaVA-v1.5-7B as both the captioner and the consistency judge**. These experiments demonstrate that even under a self-training setting, our method provides consistent improvements on the VQA task.
>
> | Method          | VQA   | MMMU  | GQA   | MM-VET | SQA   | MME  | POPE  | MMB   | MMB(CN) | VQA(v2) | LLaVA(w) | VizWiz | SEED_I | Average |
> |------------------|-------|-------|-------|--------|-------|------|-------|-------|---------|---------|----------|--------|--------|---------|
> | Self-Training   | 56.6  | **35.7**  | **63.9**  | 30.4   | **70.7**  | 1484 | 85.5  | **65.9**  | **59.3**    | **79.2**    | **69.7**     | 49.8   | **66.8**   | **62.14**   |
> | Baseline        | **58.2**  | 35.3  | 62.0  | **31.1**   | 66.8  | **1511** | **85.9**  | 64.3  | 58.3    | 78.5    | 65.4     | **50.0**   | 66.4   | 61.37   |
>
> The results from the self-training experiment demonstrate that CycleAug provides consistent improvements even when using the same model (LLaVA-v1.5-7B) as both the captioner and consistency judge. This eliminates the need for a stronger external model, showing that **the performance gains stem from the CycleAug framework itself rather than distilling knowledge from a better MLLM**.

---

> ### Author Response · Authors · 2024-12-02
>
> Dear Reviewer,
>
> If our response does not address your remaining concerns, please let us know, and we will address them promptly before the rebuttal period concludes. In addition, we have revised our manuscript based on your suggestions.
>
> Thank you!

---

### Official Review · Reviewer_Lt5H · 2024-10-30

**Soundness:** 3
**Presentation:** 3
**Contribution:** 2
**Rating:** 5
**Confidence:** 3

**Summary:**

This paper propose a novel data augmentation framework for visual instruction tuning.  The framework uses MLLM to generate detailed captions for the original images, and then generate synthetic image based on these captions. Question-answer pairs of the origin images are re-verified on the synthetic image to select the well-aligned synthetic image and question-answer triplets, thus achieving cycle-consistent visual augmentation. Experiments shows the framework can improve model performance on multiple visual question answering benchmarks without additional real data, confirming the effectiveness of the augmentation .

**Strengths:**

(1) This paper propsed a novel data augmentation framework for visual instruction tuning. By leveraging cycle-consistency sampling, the framework can generate diverse synthetic images based on existing IQA anchor triplets.
(2) The proposed method is simple and intuitive. The authors validate their method across various benchmarks, demonstrating the effectiveness of the framework.
(3) The paper is generally well written.

**Weaknesses:**

(1) Performance should be evaluated using more diverse backbones, which can better assess the robustness and effectiveness of the proposed framework.
(2) There are some works that have explored the cycle consistency in the VQA area. The approach presented in [1] involves Cycle-Consistency for Robust Visual Question Answering, wherein the model is trained not only to generate answers to given questions but also to formulate questions conditioned on specific answers. This methodology ensures that the answers predicted for the generated questions are consistent with the true answers corresponding to the original questions. This paper should clarify the relevance to the existing work [1].
[1] (CVPR 2019) Cycle-Consistency for Robust Visual Question Answering

**Questions:**

1. When generating image captions,  integrating the question-answer pair into the prompt  may improve the proportion of consistent triplets in all generated images. I hope the authors can analyze the potential trade-offs of this suggestion, such as its impact on image diversity or whether it introduces other effects into the framework.
2. Related to weakness (1). What is the impact of different MLLM and diffusion models on experimental results? I hope the author can provide additional experiments and analysis on how various MLLM and diffusion models will affect the final performance on the benchmark, and whether non-diffusion generative model architectures have any impact on the effectiveness of the framework.

---

> ### Author Response · Authors · 2024-11-30
> **Response to Weakness**
>
> We appreciate the reviewer’s constructive suggestions and apologize for the delayed response, as we have been preparing experiments to thoroughly address your concerns. In addition, we have revised our manuscript based on your suggestions.
>
> > W1:Performance should be evaluated using more diverse backbones, which can better assess the robustness and effectiveness of the proposed framework
>
> To address the reviewer’s concerns, we have conducted additional experiments across multiple MLLMs, including LLaVA-LLaMA3, LLaVA-v1.5-13B, and MGM-2B:
> | Model | $VQA^T$ | MMMU | GQA | MM-VET | SQA | MME | POPE | MMB | $MMB^{CN}$ | $VQA_{v2}$ | $LLaVA^w$ | VizWiz | $SEED_I$ | Average |
> |----------------------------|-----------|----------|----------|------------|------------|---------|----------|-----------|---------|-----------|------------|------------|----------|----------|
> | MGM-2B (w/ cycleaug)| 53.2 | 30.3 | 62.4 | 28.4 | 62.2 | 1346 | 86.1 | 60.9 | 54.4 | 78.3 | 58.4 | 50.0 | 63.5 | **58.10** |
> | MGM-2B  | 52.5 | 29.7 | 60.5 | 26.8 | 64.1 | 1330 | 85.8 | 59.0 | 48.5 | 77.4 | 56.7 | 50.0 | 62.4 | 56.91 |
> | LLaVA-13B (w/ cycleaug)| 60.1 | 38.1 | 64.2 | 35.2 | 74.1 | 1517 | 87.6 | 68.9 | 64.7 | 79.9 | 73.4 | 50.0 | 67.0 | **64.55** |
> | LLaVA-13 | 60.0 | 37.9 | 63.0 | 35.0 | 74.1 | 1503 | 86.6 | 68.2 | 63.5 | 79.6 | 71.0 | 53.6 | 66.8 | 64.18 |
> | LLaVA-LLaMA3 (w/ cycleaug)| 57.4 | 37.9 | 64.7 | 35.1 | 78.9 | 1466 | 85.0 | 73.7 | 67.9 | 79.7 | 73.6 | 50.0 | 68.9 | **65.08** |
> | LLaVA-LLaMA3 | 58.3 | 37.6 | 63.5 | 34.9 | 80.9 | 1468 | 86.5 | 71.3 | 66.2 | 79.5 | 71.4 | 50.0 | 69.6 | 64.85 |
>
> **CycleAug brings gains on more backbones**: Our method consistently outperforms baselines, with notable improvements on GQA, MMB, and MM-Vet benchmarks. Average performance gains are +1.19%, +0.37%, and +0.23% for MGM-2B, LLaVA-13B, and LLaVA-LLaMA3, respectively.
>
> > W2: This paper should clarify the relevance to the existing work [1]. [1] (CVPR 2019) Cycle-Consistency for Robust Visual Question Answering
>
> We appreciate the opportunity for further clarification. While both approaches use the term "cycle-consistency," our method differs significantly in focus and implementation:
>
> **Focus on Visual Augmentation**: Our method leverages visual foundation models, such as Stable Diffusion 3, to augment visual data, enhancing the diversity and quality of training datasets. In contrast, the work in [1] emphasizes augmenting Q&A pairs by designing a specific loss function (Eq. 1 in [1]) to explicitly enforce consistency between generated and original Q&A pairs.
>
> **Application to Large Multimodal Models**: Our approach is tailored to improve the performance of large multimodal models, comprising a vision encoder, multimodal projector, and large language model, trained using token prediction loss. In contrast, [1] focuses on validating their method across various models, which do not include large-scale multimodal architectures.

---

> ### Author Response · Authors · 2024-11-30
> **Part2: Response to Questions**
>
> > Q1: integrating the question-answer pair into the prompt may improve the proportion of consistent triplets in all generated images
>
> Thank you for the valuable suggestion regarding integrating QA pairs into the image captioning process. This approach is promising but introduces two significant challenges:
> 1. **Higher Requirements on Multi-Modal LLMs**: Generating coherent captions that effectively incorporate both QA pairs and images demands more advanced reasoning capabilities from multi-modal LLMs.
> 2. **Increased Computational Costs**: The inclusion of longer QA pairs adds more tokens to process, significantly increasing the computational overhead.
>
> To assess the effectiveness of this method, we conducted an experiment comparing image-only and QA-integrated captioning under the same amount of synthetic images. Results are as follows:
>
> | Model          | TEXT-VQA | MMMU  | GQA   | MM-VET | ScienceQA | MME     | POPE  | MMBench | MMBench-CN | VQAv2 | LLaVA(w) | VizWiz | SEED-img | Average | Time on Image Captioning (GPU*h) |
> |-----------------|----------|-------|-------|--------|-----------|---------|-------|---------|------------|-------|----------|--------|----------|---------|----------------------------------|
> | Image-Only      | 57.4     | 35.6  | 63.9  | 29.6   | 71.2      | 1506.6  | 86.4  | 67.0    | 61.2       | 79.3  | 66.5     | 50.0   | 66.9     | 62.33   | 134                               |
> | QA-Integrated   | 57.37    | 35.9 | 64.0 | 30.2  | 71.4     | 1501.7 | 86.0 | 67.0   | 61.2      | 79.3 | 63.8    | 50.0  | 67.1    | 62.18   | 447                              |
>
> **Performance**: QA-integrated captioning shows marginal improvements in some benchmarks (e.g., MMMU, GQA, and SEED-img) but slightly lower results in others (e.g., MME). The overall average remains comparable (62.18 vs. 62.33).
>
> **Computational Cost**: The time required for QA-integrated captioning is significantly higher, increasing from **66 GPU-hours** to **254 GPU-hours**, reflecting the additional processing of QA pairs.
>
> **Conclusion**: While QA-integrated captioning may slightly enhance alignment between captions and QA pairs, the marginal performance gains do not justify the steep increase in computational costs under the current framework. Future advancements in computationally efficient multi-modal models may make this approach more viable.
> > Q2: additional experiments and analysis on how various MLLM and diffusion models will affect the final performance on the benchmark
>
> Thank you for the constructive suggestion. As highlighted in our response to W1, our pipeline demonstrates adaptability to various architectures, parameter scales, and visual encoders. To further extend our analysis, we are currently experimenting with autoregressive-based model, VAR[1]. Such experiment will help clarify how diverse generative models impact the effectiveness and performance of our framework across benchmarks. We will provide detailed results and analysis in the updated manuscript.
>
> [1] Visual autoregressive modeling:Scalable image generation via next-scale prediction. 2024.

---

> > ### Comment · Reviewer_Lt5H · 2024-12-02
> >
> > Thanks for clarifying and the additional experiment result. I appreciate the author's responses in addressing my concerns and have carefully read the discussions with other reviewers.
> > The proposed approach itself may seem somewhat incremental, since my current score already takes these various aspects into account, I will maintain my score.

---

> ### Author Response · Authors · 2024-12-02
>
> Dear Reviewer,
>
> If our response does not address your remaining concerns, please let us know, and we will address them promptly before the rebuttal period concludes. In addition, we have revised our manuscript based on your suggestions.
>
> Thank you!

---

### Official Review · Reviewer_rJUF · 2024-11-01

**Soundness:** 3
**Presentation:** 3
**Contribution:** 3
**Rating:** 6
**Confidence:** 3

**Summary:**

This paper presents CYCLEAUG, a data augmentation framework for MLLMs that generates diverse, cycle-consistent synthetic image-question-answer triplets. By ensuring alignment between generated images and their corresponding QA pairs, CYCLEAUG enhances model robustness and generalization without requiring additional real-world data. Experiments verify the effectiveness of the proposed methods.

**Strengths:**

1. The idea of using cycle-consistency for MLLMs for data augmentation is novel.  While data augmentation and synthetic data generation are established fields, CYCLEAUG advances these techniques by ensuring semantic alignment between generated images and their question-answer pairs, effectively addressing quality control and enhancing robustness in synthetic data generation.

2. The paper is well-organized and easy to follow.

3. The proposed method can be seamlessly integrated with existing visual instruction tuning frameworks, and its modular design holds potential for adaptation and extension to a broader range of multimodal tasks.

**Weaknesses:**

1. The author does not provide a detailed discussion on the specific computational costs and resource consumption of the proposed method.  This could pose challenges for actual application environments with limited resources.

2. The paper presents a method for creating diverse synthetic images from multi-turn question-answer dialogues but lacks an in-depth exploration of alternative dialogue generation strategies, which could restrict the analysis of image diversity and relevance, impacting the overall model performance.

3. In addition to the direct sampling method, there are numerous data augmentation techniques applicable to the field of multimodal learning.  Incorporating comparative experiments would aid in a more comprehensive assessment of CYCLEAUG's performance and potentially reveal its advantages.

**Questions:**

Refer to Weakness.

---

> ### Author Response · Authors · 2024-11-30
> **Part 1: Response to Weakness 1**
>
> We appreciate the reviewer’s constructive suggestions and apologize for the delayed response, as we have been preparing experiments to thoroughly address your concerns. In addition, we have revised our manuscript based on your suggestions.
>
> > W1: a detailed discussion on the specific computational costs and resource consumption of the proposed method
>
> We appreciate the reviewer’s suggestion to report the computational cost and filtering proportions, which we have now calculated. Below, we detail the FLOPs, latency, and overall time for each stage of our process, along with the proportion of images filtered out by cycle-consistency sampling:
>
> **Cost Analysis**:
> The computational cost is estimated based on the FLOPs and latency of each component, averaged across 200 random samples, and scaled to the total dataset size. The measurements are as follows:
> | Model                          | FLOPS (T)  | Latency (s/sample) | # Images/QA | Overall Time (s) | Overall Time (h·GPU) |
> |--------------------------------|------------|---------------------|-------------|------------------|-----------------------|
> | SD3 (Image Generation)          | 87.9       | 4.1                 | 117,576     | 482,062          | 133.9                 |
> | VAR-d-30[1] (Image Generation)   | 2.3        | 0.7                 | 117,576     | 82,303           | 22.9                   |
> | LLaVA-LLaMA-3 (Captioning & VQA) | 8.3       | 1.2                 | 273,144     | 327,773          | 91.0                  |
> | SBERT Model (Consistency Judge) | 0.2        | 0.0                 | 273,144     | 13,250           | 3.7                   |
>
> [1] Visual autoregressive modeling:Scalable image generation via next-scale prediction. 2024.
>
> **Filtering Proportion**:
> Out of the 273,144 images initially generated, **117,576 images (43%)** were retained after applying cycle-consistency sampling. This step effectively eliminates 57% of the images, ensuring the final dataset is both diverse and of high quality.
>
> **FLOPs and time-cost distribution**:
> We measure the FLOPs and time cost of CycleAug to evaluate its computational efficiency. As shown in Table, the image generation step (Stable Diffusion 3) accounts for the highest computational cost, requiring 133.9 GPU-hours due to its high FLOPs (87.9T) and latency (4.1s/sample). Image captioning and VQA (LLaVA-Llama-3) strike a balance, processing 273,144 samples in 91.0 GPU-hours, with moderate FLOPs (8.3T) and latency (1.2s/sample). The consistency judgment step (Sbert-Model) is the most efficient, contributing only 3.7 GPU-hours.This analysis highlights that image generation is the main bottleneck, while captioning and consistency judgment remain computationally efficient. Optimizing the generation step could significantly reduce overall cost, improving scalability for larger datasets.
>
> **Large Room for FLOPs reduction**:
> Although the FLOPs and time costs are significant for Stable Diffusion 3, we also measured the performance of VAR, an autoregressive-based text-to-image generation method with comparable quality to diffusion-based methods (FID: 1.8 vs. 1.56). VAR requires only 22.9 GPU hours to generate 117k images, making it both efficient and scalable to larger datasets.
>
> Since our current focus is on demonstrating the proof of concept, we have not emphasized efficiency. However, **the performance of VAR highlights the potential for our pipeline to achieve high efficiency**, especially when integrated with more computationally efficient models.
>
> **Conclusion**:
> CycleAug's computational cost is significant, primarily due to the image generation step, but the filtering process ensures high-quality data critical for performance improvements. Efficient alternatives like VAR, with comparable quality and lower costs, offer potential for optimization, making CycleAug more scalable for larger datasets.

---

> ### Author Response · Authors · 2024-11-30
> **Part 2: Response to Weakness 2 & 3**
>
> > W2: lacks an in-depth exploration of alternative dialogue generation strategies,which could restrict the analysis of image diversity and relevance, impacting the overall model performance.
>
> We appreciate the reviewer’s comment highlighting the potential impact of exploring alternative dialogue generation strategies. Our method focuses on generating diverse images by captioning real images using a VLM-captioner. Since it is not feasible for the VLM-captioner to capture every detail of an image, the Stable Diffusion 3 (SD3) model synthesizes high-fidelity images based on its own distribution. This process inherently introduces diversity, effectively serving as data augmentation.
>
> To ensure alignment between the synthesized images and the QA pairs, our method employs a consistency judge. This guarantees both diversity and relevance, resulting in an effective data augmentation strategy for improving model performance.
>
> While dialogue generation strategies are not our focus, we agree that such strategies have the potential to further enhance data augmentation.
>
> > W3:In addition to the direct sampling method, there are numerous data augmentation techniques applicable to the field of multimodal learning.
>
> Thank you for highlighting the importance of comparative experiments with other augmentation techniques. We agree that incorporating such comparisons is valuable for assessing CycleAug's performance and showcasing its potential advantages.
>
> We conducted experiments with transformation that composes of commonly used augmentation techniques including random cropping and resizing, rotation, and color jittering. The results are summarized below:
>
> | Model      | TEXT-VQA | MMMU  | GQA   | MM-VET | ScienceQA | MME     | POPE  | MMBench | MMBench-CN | VQAv2 | LLaVA(w) | VizWiz | SEED-img | Average |
> |------------|----------|-------|-------|--------|-----------|---------|-------|---------|------------|-------|----------|--------|----------|---------|
> | Composed   | 57.5     | 34.3  | 63.2  | 28.8   | 70.8      | 1491    | 85.6  | 65.6    | 60.5       | 79.1  | 64.5     | 48.5   | 66.1     | 61.46   |
> | CycleAug   | 57.4     | 35.6  | 63.9  | 29.6   | 71.2      | 1506.6  | 86.4  | 67.0    | 61.2       | 79.3  | 66.5     | 50.0   | 66.9     | 62.33   |
>
> Such composed augmentation diversify the visual appearance of images but negatively impact concept alignment with QA pairs. For instance: **Rotation** can alter the spatial relationships in images, which are crucial for tasks requiring positional reasoning. **CycleAug**, in contrast, our method achieves higher alignment between images and QA pairs by leveraging cycle-consistency sampling and maintaining the semantic coherence of the data.
>
> **Conclusion**: The comparative experiments demonstrate that CycleAug outperforms traditional augmentation techniques by preserving concept alignment while diversifying the data. These results highlight the importance of maintaining semantic consistency in multimodal learning tasks.

---

> ### Author Response · Authors · 2024-12-02
>
> Dear Reviewer,
>
> If our response does not address your remaining concerns, please let us know, and we will address them promptly before the rebuttal period concludes. In addition, we have revised our manuscript based on your suggestions.
>
> Thank you!

---

> > ### Comment · Reviewer_rJUF · 2024-12-02
> >
> > Thanks for your responses. Most of my concerns have been addressed and I would like to maintain my original rating.

---

### Official Review · Reviewer_ZGm1 · 2024-11-01

**Soundness:** 3
**Presentation:** 3
**Contribution:** 2
**Rating:** 6
**Confidence:** 4

**Summary:**

The paper presents a data augmentation framework called CYCLEAUG, which generates multiple images paired with existing QA pairs using Stable Diffusion. Several post-processing mechanisms are employed to ensure consistency between the generated images and the anchor QA pairs. Experimental results demonstrate slight improved performance of MLLMs when incorporating this augmented data for VQA tasks.

**Strengths:**

1. The paper is well written and organized.
2. The method is simple and easy to follow and replicate.

**Weaknesses:**

1. The main issue is that the paper aims to address the limitations of images generated by SD, specifically "homogeneous content" and the "can not handle negation word" (lines 253-257), but the work does not have any concrete efforts to resolve these problems.
2. The method is not well-suited for text-rich VQA, as the current SD model struggles with generating characters.
3. The performance gain is quite limited; for example, in Table 3, the augmented images even have a negative impact on the VQAT dataset.
4. The paper would benefit from reporting the cost (e.g., FLOPS, latency) required to produce these images, as well as the proportion of images filtered out by cycle consistency sampling.

**Questions:**

1. Besides using detailed LLM-generated captions to prompt SD, have you tried generating an image with SD where the starting point is a noisy anchor image? I believe this could also help maintain consistency with the QA pairs.

---

> ### Author Response · Authors · 2024-11-30
> **Part 1: Response to Weakness 1-3**
>
> We appreciate the reviewer’s constructive suggestions and apologize for the delayed response, as we have been preparing experiments to thoroughly address your concerns. In addition, we have revised our manuscript based on your suggestions.
>
> > W1: concrete efforts to resolve *homogeneous content* and the *can not handle negation word*:
>
> We appreciate the opporunity to clarify the issues of (1) homogeneous content and (2) the inability to handle negation words in image generation. The two issues primarily arise in direct sampling because the generated images are synthesized solely based on brief QA pairs. This brevity often lacks the detailed information needed to produce rich and diverse content, resulting in homogeneous outputs. While negation-related challenges could be solved by negative prompt for (2), the issue of homogeneous content (1) persists and may even be exacerbated.
> **To address these limitations, our work adopts a “complete captioning to synthesizing” framework**. This approach involves leveraging LLMs to generate detailed and context-rich captions from images. The enhanced captions mitigate the issue of homogeneous content by introducing more diverse and descriptive information. Moreover, this framework inherently resolves the negation problem, as the captioning process guided by LLMs naturally converts complex or negated expressions into straightforward and explicit descriptions.
>
> In summary, we are fully aware of the concerns raised and we emphasize that our proposed cycle-consistency sampling framework specifically targets and resolves these issues by enhancing image diversity and addressing negation through a robust captioning pipeline.
>
> > W 2&3: The performance gain is quite limited; for example, in Table 3, the augmented images even have a negative impact on the VQAT dataset.
>
> We appreciate the reviewer’s observation on text-rich VQA tasks. This stems from the common limitations of the Stable Diffusion models in generating characters, which hinders its performance on text-focused QA tasks like TextVQA.
> However, we would like to highlight that the majority of our visual augmentation is applied to COCO and GQA datasets, which focus on broader visual reasoning tasks. The efficacy of our augmentation is supported by improvements in accuracy on related benchmarks, including GQA (+1.9%), MMB (+2.7%),MMB-CN(+2.9%) and MMB-CN (+0.5%). These results demonstrate that our method effectively enhances performance on major distributions, even though its capability for text-related QA tasks remains limited due to the text-rendering constraints of Stable Diffusion 3.
> We also note that the limitations with text-rich VQA tasks have the potential to be addressed with the advance of image generation models (such as [ideogram](https://ideogram.ai/) and [Text-Diffuser2](https://arxiv.org/pdf/2311.16465)), paving the way for better support of text-rich scenarios in the future.

---

> ### Author Response · Authors · 2024-11-30
> **Part2: Response to Weakness 4 &  Question 1**
>
> > W4: The paper would benefit from reporting the cost (e.g., FLOPS, latency) required to produce these images, as well as the proportion of images filtered out by cycle consistency sampling.
> We appreciate the reviewer’s suggestion to report the computational cost and filtering proportions, which we have now calculated. Below, we detail the FLOPs, latency, and overall time for each stage of our process, along with the proportion of images filtered out by cycle-consistency sampling:
>
> **Cost Analysis**:
> The computational cost is estimated based on the FLOPs and latency of each component, averaged across 200 random samples, and scaled to the total dataset size. The measurements are as follows:
> | Model                          | FLOPS (T)  | Latency (s/sample) | # Images/QA | Overall Time (s) | Overall Time (h·GPU) |
> |--------------------------------|------------|---------------------|-------------|------------------|-----------------------|
> | SD3 (Image Generation)          | 87.9       | 4.1                 | 117,576     | 482,062          | 133.9                 |
> | VAR-d-30[1] (Image Generation)   | 2.3        | 0.7                 | 117,576     | 82,303           | 22.9                   |
> | LLaVA-LLaMA-3 (Captioning & VQA) | 8.3       | 1.2                 | 273,144     | 327,773          | 91.0                  |
> | SBERT Model (Consistency Judge) | 0.2        | 0.0                 | 273,144     | 13,250           | 3.7                   |
>
> [1] Visual autoregressive modeling:Scalable image generation via next-scale prediction. 2024.
>
> **Filtering Proportion**:
> Out of the 273,144 images initially generated, **117,576 images (43%)** were retained after applying cycle-consistency sampling. This step effectively eliminates 57% of the images, ensuring the final dataset is both diverse and of high quality.
>
> **FLOPs and time-cost distribution**:
> We measure the FLOPs and time cost of CycleAug to evaluate its computational efficiency. As shown in Table, the image generation step (Stable Diffusion 3) accounts for the highest computational cost, requiring 133.9 GPU-hours due to its high FLOPs (87.9T) and latency (4.1s/sample). Image captioning and VQA (LLaVA-Llama-3) strike a balance, processing 273,144 samples in 91.0 GPU-hours, with moderate FLOPs (8.3T) and latency (1.2s/sample). The consistency judgment step (Sbert-Model) is the most efficient, contributing only 3.7 GPU-hours.This analysis highlights that image generation is the main bottleneck, while captioning and consistency judgment remain computationally efficient. Optimizing the generation step could significantly reduce overall cost, improving scalability for larger datasets.
>
> **Large Room for FLOPs reduction**:
> Although the FLOPs and time costs are significant for Stable Diffusion 3, we also measured the performance of VAR, an autoregressive-based text-to-image generation method with comparable quality to diffusion-based methods (FID: 1.8 vs. 1.56). VAR requires only 22.9 GPU hours to generate 117k images, making it both efficient and scalable to larger datasets.
>
> Since our current focus is on demonstrating the proof of concept, we have not emphasized efficiency. However, **the performance of VAR highlights the potential for our pipeline to achieve high efficiency**, especially when integrated with more computationally efficient models.
>
> **Conclusion**:
> CycleAug's computational cost is significant, primarily due to the image generation step, but the filtering process ensures high-quality data critical for performance improvements. Efficient alternatives like VAR, with comparable quality and lower costs, offer potential for optimization, making CycleAug more scalable for larger datasets.
>
> > Q1: generating an image with SD where the starting point is a noisy anchor image
>
> We thank for the constructive suggestion. Starting from a noisy anchor image is indeed an intriguing idea and holds potential for improving consistency with QA pairs. However, implementing this approach is nontrivial, as Stable Diffusion denoises in the latent feature space rather than the raw pixel space. This difference adds complexity to initializing the process with a noisy anchor image.
>
> While this idea is promising, we plan to explore it in future work to address the technical challenges and assess its impact on maintaining consistency and improving generation quality. Thank you again for the insightful suggestion.

---

> ### Author Response · Authors · 2024-12-02
>
> Dear Reviewer,
>
> If our response does not address your remaining concerns, please let us know, and we will address them promptly before the rebuttal period concludes. In addition, we have revised our manuscript based on your suggestions.
>
> Thank you!

---

### Meta-Review · Area_Chair_f6ge · 2024-12-21

**Metareview:**

The paper presents a data augmentation framework called CYCLEAUG, which generates multiple images paired with existing QA pairs using Stable Diffusion. Several post-processing mechanisms are employed to ensure consistency between the generated images and the anchor QA pairs. CYCLEAUG enhances model robustness and generalization without requiring additional real-world data. Experimental results demonstrate slightly improved performance of MLLMs when incorporating this augmented data for VQA tasks.

Strengths:
+ The paper is well-written and organized.
+ The method is simple and intuitive.
+ The authors validate their method across various benchmarks, demonstrating the effectiveness of the framework.

Weaknesses:
+ The novelty of the proposed method is incremental.
+ The experimental setting is not reasonable, it uses a larger MLLM to generate training data, and utilizes it to fine-tune a smaller MLLM.
+ More data augmentation techniques on multimodal learning should be compared as baselines.
+ More ablation studies (such as more diverse backbones, training/inference computational cost) are needed.

**Additional Comments On Reviewer Discussion:**

After the rebuttal, this submission received mixed reviews (35566). The main remaining concern is that: the novelty of the proposed data augmentation is limited. As suggested by the reviewers, it should add more comprehensive and detailed discussions and comparisons with other similar cycle-consistency data augmentation techniques. Thus, I recommend Reject for this submission version.

---

### Decision · Program_Chairs · 2025-01-22

Reject